# Parallel encoding of speech in human frontal and temporal lobes

Patrick W. Hullett[1,2], Matthew K. Leonard [2,3], Maria Luisa Gorno-Tempini[1,2], Maria Luisa Mandelli[1,2] & Edward F. Chang [2,3] ✉

Models of speech perception are centered around a hierarchy in which auditory representations in the thalamus propagate to primary auditory cortex, then to the lateral temporal cortex, and finally through lateral cortical areas to reach the frontal lobe. However, it is unclear if speech-evoked activity in frontal cortex strictly reflects downstream processing from this hierarchical pathway or whether there are long-range parallel connections from the low-level areas (thalamus, primary auditory cortex) to the frontal lobe. Here, we used high-density direct cortical recordings, high-resolution diffusion tractography, and hemodynamic functional connectivity to evaluate for evidence of direct parallel inputs to frontal cortex from low-level areas. We found that neural populations in the frontal lobe show speech-evoked responses that are synchronous with the shortest-latency responses in the superior temporal gyrus (STG) and encode spectrotemporal speech content indistinguishable from spectrotemporal encoding patterns observed in the STG. Additionally, we find white matter tractography and functional connectivity patterns that connect the auditory nucleus of the thalamus and the primary auditory cortex to the frontal lobe. Together, these results support the existence of robust long-range parallel inputs from low-level auditory areas to apical areas in the frontal lobe of the human speech network.

To understand how the brain performs remarkable computational feats like speech perception, it is necessary to understand the cortical regions involved and the network architecture that connects them. The speech network in the human brain is postulated to have a largely hierarchical organization with information flow from subcortical structures to primary auditory cortex, then to lateral temporal cortex through successive dorsal and ventral pathway areas to reach apical targets in the frontal lobe[1,2]. Thus, activity in the frontal lobe during speech perception is thought to reflect downstream computational processes inherited from successive stimulus transformations in areas along the dorsal and ventral lateral cortex pathways (Fig. 1A, purple).

In addition to this hierarchical structure of the speech network, within nearby levels, there are short-range, local, parallel connections[3,4]. However, long-range parallel connections from the bottom of the network (i.e., sensory nuclei in the thalamus or primary auditory cortex) to apical regions in the frontal lobe have yet to be described (Fig. 1A, orange arrows). In support of such long-range parallel pathways, studies in primates found evidence for frontal lobe auditory responses that have short latencies in the range of those observed in lateral temporal cortex[5,6]. Additionally, auditory activity in these areas showed evidence of spectrotemporal representations[7]. These data, although in primates, raise the possibility of frontal cortex responses that may reflect parallel inputs from the auditory thalamus

[1]Department of Neurology, University of California, San Francisco, San Francisco, CA, USA. [2]Weill Institute for Neurosciences, University of California, San Francisco, CA, USA. [3]Department of Neurological Surgery, University of California, San Francisco, San Francisco, CA, USA. ✉e-mail: edward.chang@ucsf.edu

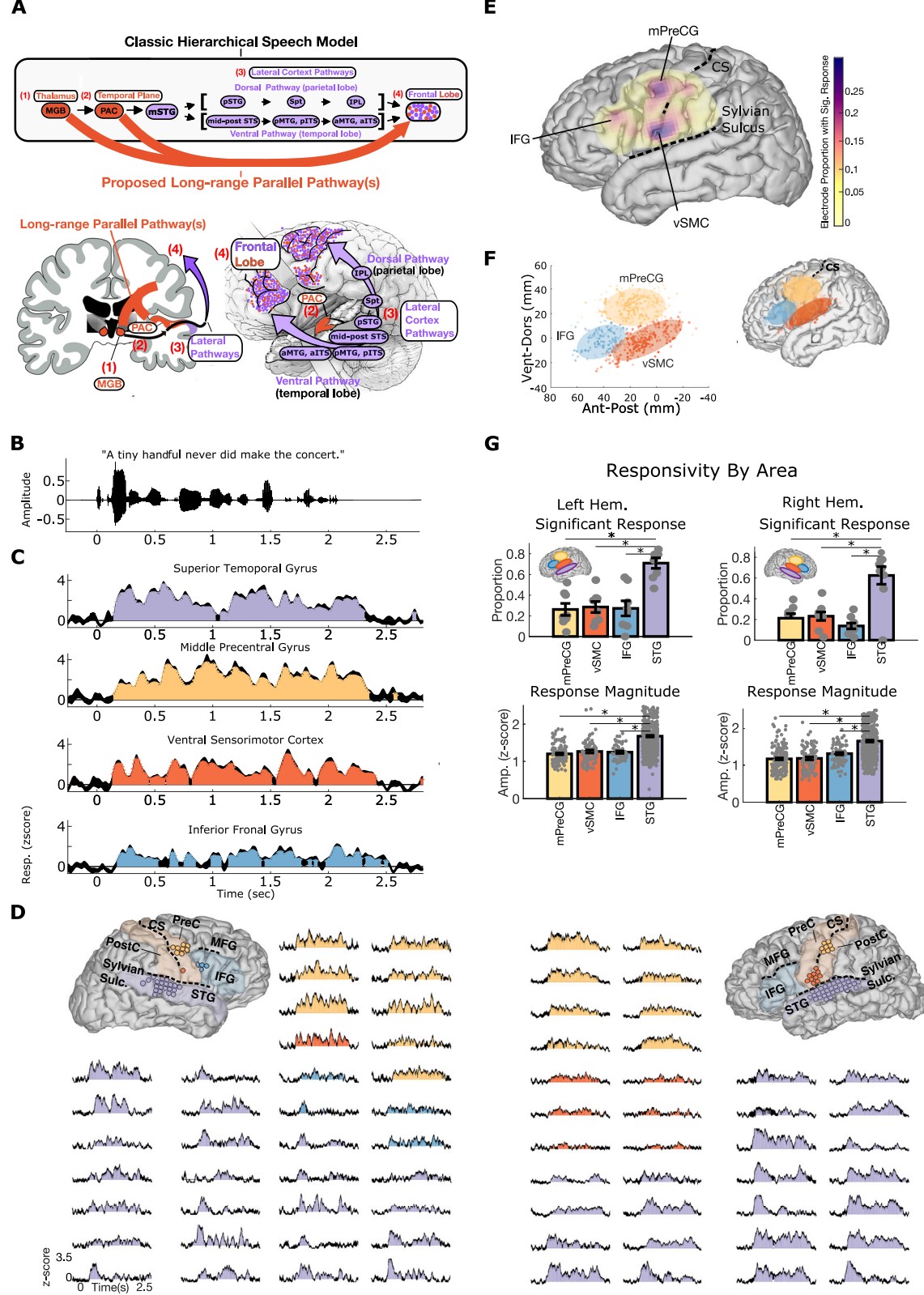

and primary auditory cortex, forming a long-range parallel pathway to the frontal cortex.

To test this hypothesis, we used electrocorticography (ECoG) to identify speech responses in the frontal and temporal lobes in awake participants while they passively listened to natural speech. We specifically asked whether there is evidence for: (1) short-latency speech-evoked responses that are synchronous between the frontal lobe and

the earliest latency responses in the superior temporal gyrus (STG) and (2) whether such activity is explained by spectrotemporal encoding models as would be expected for direct inputs from low-level areas such as the primary auditory cortex or the medial geniculate body of the thalamus. Here, we identify three areas in the frontal lobe cortex that are activated by speech with latencies that occur simultaneously or even before the earliest latencies in the superior temporal gyrus

**Fig. 1 | Suprasylvian speech response characteristics. A** The classic hierarchical speech pathway from the medial geniculate body (MGB) of the thalamus to the front lobe is shown in the top panel. The orange arrows illustrate the hypothesized long-range parallel pathways from the medial geniculate body (MGB) and primary auditory cortex (PAC) to the frontal lobe. **B** Acoustic waveform of a single example sentence presented to participants. **C** Example responses to a single sentence from the right hemisphere participant in (**D**). **D** Representative example of two participants showing example responses in suprasylvian cortex and superior temporal gyrus (STG) to the single sentence shown in (**B**). The brain reconstructions show the spatial locations of responses to this sentence. This shows high similarity in response characteristics in the frontal lobe and STG. **E** Spatial distribution of significant responses to speech across all participants. This shows three areas of peak response probability in IFG, middle precentral gyrus (mPreCG), and ventral sensorimotor cortex (vSMC) (electrodes collapsed to the left hemisphere). **F** Spatial clustering using a Gaussian mixture model demonstrates three areas of speech responses in IFG, mPreCG, and vSMC. The number of clusters was determined by Bayesian information criterion and silhouette criterion values (both converged on 3

clusters)[13]. **G** Suprasylvian area versus STG responsivity. The proportion of electrodes with significant responses and the mean response amplitude is lower in IFG, mPreCG, and vSMC than in STG ($p < 0.005$, two-sided Wilcoxon rank-sum test, Bonferroni corrected, n proportion samples for left/right hem. = 9/8 IFG, 8/8 mPreCG, 9/8 vSMC, 9/8 STG, n response amplitudes for left/right hem. = 52/84 IFG, 120/180 mPreCG, 98/91 vSMC, 645/548 STG). Each area has no left-right hemispheric differences regarding the proportion or magnitude of significant responses ($p > 0.05$, two-sided Wilcoxon rank-sum test). All data are presented as mean values ± SEM. CS central sulcus; Hem hemisphere; IFG inferior frontal gyrus; IPL inferior parietal lobe; ITS inferior temporal sulcus; MGB medial geniculate body; mPreCG middle precentral gyrus; MTG middle temporal gyrus; PostC postcentral gyrus; PreC precentral gyrus; Spt Sylvian fissure at the parieto-temporal boundary; STG superior temporal gyrus; STS superior temporal sulcus; Sulc. sulcus; vSMC ventral sensorimotor cortex. **A** contains brain illustrations from Kenneth Probst, reproduced with permission, licensed under a Creative Commons Attribution 4.0 https://creativecommons.org/licenses/by/4.0/deed.en.

(STG). Additionally, neural representations in these short-latency populations encode spectrotemporal content that is indistinguishable from populations in STG. Finally, to further establish evidence for anatomic connections that reflect parallel inputs from lower-level areas to the frontal lobe, we use white matter tractography and functional connectivity and find connections between frontal lobe regions and the auditory nucleus of the thalamus (MGB) and primary auditory cortex. Overall, these results demonstrate a fundamental divergence in the hierarchical architecture that is usually assumed to underlie cortical speech processing and shows multiple lines of evidence for long-range parallel inputs from low-level areas to frontal lobe cortical regions in the human speech network.

## Results

### Frontal lobe areas are activated by passive speech listening

To characterize the extent to which neural populations in frontal lobe respond to speech, ECoG participants passively listened to 10–40 min of natural speech, which consisted of prerecorded sentences (2–4 s each) from the phonetically transcribed TIMIT speech corpus[8]. We extracted activity in the high-gamma (70–150 Hz) range[9], which correlates with spiking activity[10], spike-based tuning properties[11], and reflects contributions from single neurons throughout the cortical depth[12]. At single electrodes, average responses to an example sentence (Fig. 1B) were highly robust and extended through the duration of the sentence in STG and three areas in the frontal lobe, including middle precentral gyrus (mPreCG; Fig. 1C; yellow), ventral sensorimotor cortex (vSMC; Fig. 1C; orange), and inferior frontal gyrus (IFG; Fig. 1C; blue). Figure 1D shows two example participants that illustrate typical electrode coverage in the left and right hemispheres and responses to a single sentence of speech that are qualitatively similar to responses in STG.

Across all 17 participants, we quantified the spatial distribution of responses to speech in the frontal and parietal cortex. We observed significant responses throughout the areas covered by ECoG grids, with three peaks of responsiveness in mPreCG, vSMC, and IFG (Fig. 1E, significant responses defined as electrodes with $p < 0.05$ for any response time bin, Bonferroni corrected for the number of time bins in each sentence, Wilcoxon rank-sum test). To further test whether these three peaks reflect spatially clustered responses, we used Gaussian mixture modeling to cluster the electrodes. The three areas that emerged reflect clusters of electrodes in mPreCG, vSMC, and IFG (Fig. 1F; cluster number = 3 determined using both Bayesian information criterion, $p < 0.05$, and Silhouette criterion[13] $p < 0.05$). Thus, in addition to STG, there are three regions within the frontal lobe cortex that have significant responses to natural speech.

We compared the basic response properties in these three suprasylvian areas to STG, which is known to be a critical area that is

central to speech perception[14,15]. To characterize the overall responsivity, we first quantified the proportion of all electrodes within an area that showed a significant response to speech (Fig. 1G, top row). Although all three regions had electrodes with significant speech-evoked activity in both hemispheres, there was a lower proportion of speech-responsive electrodes in frontal lobe areas compared to the proportion of speech-responsive electrodes in STG ($p < 0.005$, two-sided Wilcoxon rank-sum test, Bonferroni corrected; Fig. 1G, top row). Additionally, while significantly greater than zero, the average response magnitude in suprasylvian areas was lower than STG ($p < 0.005$, two-sided Wilcoxon rank-sum test, Bonferroni corrected; Fig. 1G, bottom row). For both the proportion of significant electrodes and the response magnitude, there were no significant differences between hemispheres ($p > 0.05$; two-sided Wilcoxon rank-sum test), suggesting that frontal lobe speech-evoked activity is similar between right and left hemispheres[16,17].

### Frontal lobe areas have short-onset latencies like the superior temporal gyrus

Having established the existence of robust speech-evoked activity throughout bilateral frontal lobe cortex during passive listening, we asked whether any of these neural populations had short-onset latencies that are synchronous or faster than the earliest onset latencies in lateral temporal cortex. If present, this would be consistent with parallel pathways from lower-order auditory areas to frontal and temporal lobes. To address this question, we were careful to only select participants with extensive coverage across STG and these frontal lobe areas within a single participant (Supplementary Fig. 1). The seventeen participants included in this study is a subset of all participants with grid-based coverage over a 10-year span in our center. Additionally, for higher spatial resolution, all participants have specialized high-density recording grids with 4 mm inter-electrode distance in contrast to 10 mm inter-electrode distance used at other centers[18]. Direct visual examination of speech-evoked activity showed near-synchronous shortest latency responses across frontal areas and STG (Fig. 2A, B, D). Furthermore, there were within-participant examples of frontal electrodes with onset latencies synchronous or earlier than the shortest onset latencies in the temporal lobe (Fig. 2C). To quantify this, we calculated the response onset latencies across all participants, similar to prior characterizations in humans and primates[19,20] (Fig. 2A, D). This showed electrodes throughout the temporal, frontal, and parietal lobes with short response latencies, often less than 80 ms.

For each region, we quantified the latency distribution, which showed similar short-latency responses across all areas (Fig. 2D). To test for significant differences in short-onset latencies between areas, we used a temporal cutoff to define the left-sided tail of each

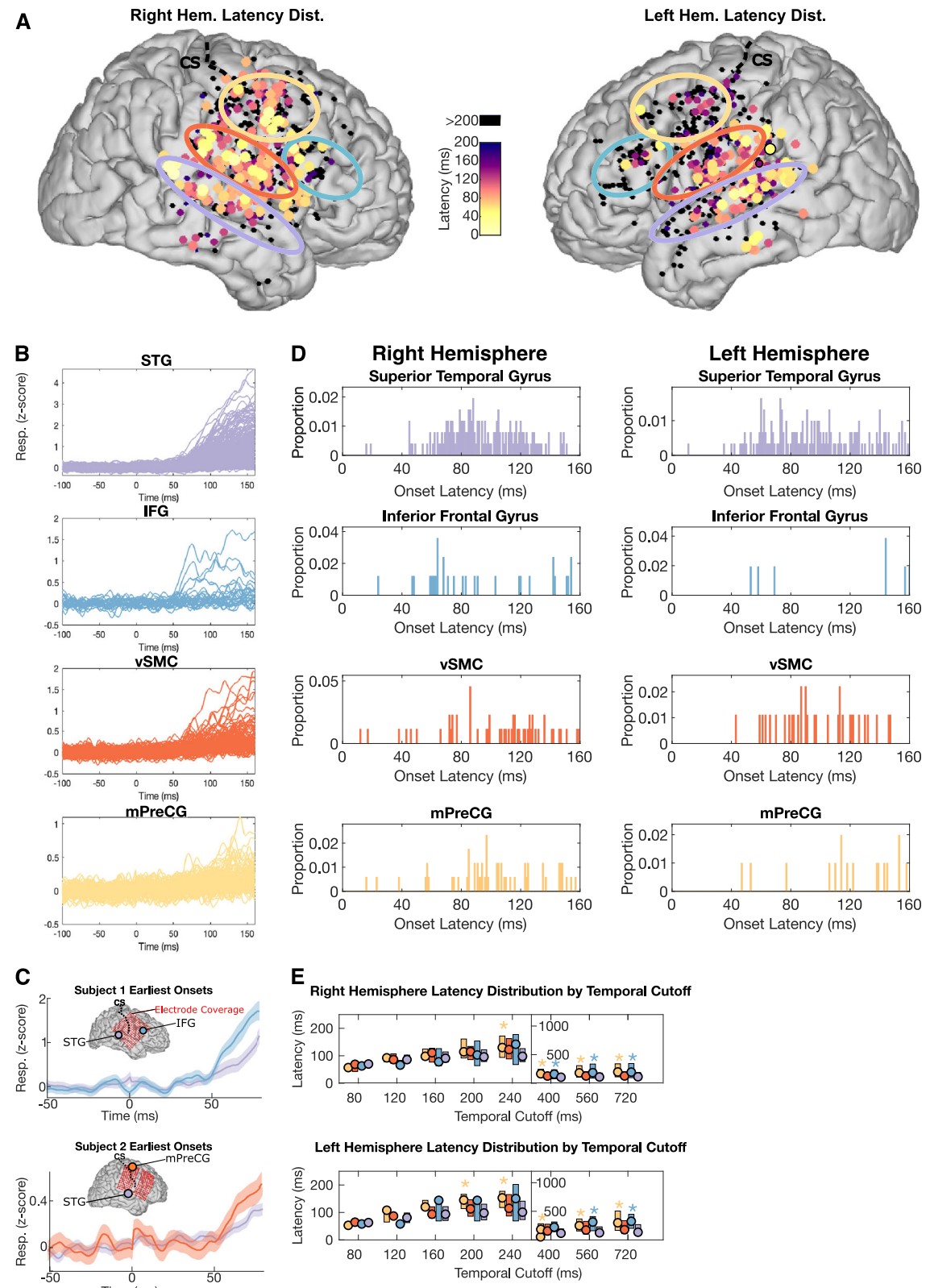

distribution as the short-onset latencies of interest (Fig. 2E). The temporal cutoff ranged from 80 to 720 ms, with significance testing at each cutoff. As shown, onset latencies were not significantly different between frontal lobe areas (IFG, mPreCG, vSMC) and STG for latencies up to 200 ms in the left hemisphere and 240 ms in the right hemisphere (Fig. 2E, *p* > 0.05, two-sample Kolmogorov–Smirnov test, Bonferroni corrected). This data shows short-onset latencies within the same hemisphere are not significantly different between frontal lobe

areas and STG. Thus, in response to natural speech, IFG, mPreCG, and vSMC are active beginning at the same time as neural populations in STG, consistent with parallel inputs to STG and frontal lobe areas.

### Short-onset latency inputs to the frontal and temporal lobe encode the same spectrotemporal speech information

A feature of most functional-anatomical models of speech is frontal lobe areas receive inputs from the temporal lobe through dorsal and

**Fig. 2 | Frontal lobe areas have short-onset latencies like the superior temporal gyrus. A** Spatial map of response latencies show short-onset latencies in STG, IFG, mPreCG, and vSMC (onset defined as first 1-ms time bin in which the response *p*-value is <0.05 for 15 consecutive 1-ms bins[19,20]; Wilcoxon rank-sum test). **B** All responses across participants with onset latencies less than 200 ms. **C** Two example participants showing the electrode sites with shortest onset latency in the STG and frontal lobe. As shown, the earliest onset latencies within a subject are similar between the frontal lobe and STG. **D** Latency distribution by area for onset latencies less than 160 ms (the proportion y-axis is relative to the full distribution of latencies up to 1600 ms). This shows the earliest onset latencies are similar across areas and is consistent with low-latency parallel inputs to each area. **E** Latency distribution boxplots for all onset latencies up to the temporal cutoff specified on the x-axis (80–720 ms). There is no significant difference between STG and IFG, mPreCG, or vSMC up to 240 ms in the right hemisphere and 200 ms in the left hemisphere (* = *p* < 0.05, two-sample Kolmogorov–Smirnov test, Bonferroni corrected, n latencies for left/right hem. = 52/84 IFG, 120/180 mPreCG, 98/91 vSMC, 330/274 STG). All box plots show the median (circle), 25th and 75th percentiles (box). These indistinguishable short-onset latency times are consistent with a subset of parallel inputs to the frontal cortex and STG rather than strictly hierarchical inputs from STG to the frontal cortex. CS central sulcus; IFG inferior frontal gyrus; mPreCG middle precentral gyrus; STG superior temporal gyrus; vSMC ventral sensorimotor cortex.

ventral pathways and may reflect qualitatively different (and perhaps higher-order) representations of speech[1,2]. Given the existence of parallel short-latency responses in the frontal and temporal lobes, we asked if the spectrotemporal information encoded in these responses fundamentally differed between lobes. To test this, we computed spectrotemporal receptive fields (STRFs, Fig. 3A) for short-onset latency sites (<200 ms) in STG, IFG, mPreCG, and vSMC. In all three frontal lobe areas, responses to speech were well-predicted by STRFs (mean r = 0.39 ± 24, Fig. 3B), with some electrodes reaching r = 0.75, demonstrating that frontal lobe neural population activity is well-explained by spectrotemporal models.

Next, we asked whether the spectrotemporal content encoded in short-onset (<200 ms) frontal lobe electrodes is similar to STG. Based on visual examination of short-onset STRFs from each frontal lobe area and STG, there was a high degree of similarity in spectrotemporal encoding (Fig. 3C). Additionally, the STRF features correlated with spectrotemporal structure seen in phonetic features similar to STG[21]. For example, the bottom row of Fig. 3C shows broadband excitatory tuning followed shortly by broadband inhibitory tuning. This is the canonical spectrotemporal structure that is tuned for plosives in phonetic feature space[21]. To quantify this we examined the STRF weights and computed three key parameters (Fig. 3D): (1) the best frequency (first column)[22,23], (2) spectral tuning (second column)[24], and (3) temporal tuning (third column)[24,25]. For each of these parameters, there was no significant difference across regions (*p* > 0.05 Kruskal–Wallis test and cosine similarity permutation test of mean spectral and temporal tuning vectors[26]).

While individual STRF tuning properties were not different between STG and frontal lobe areas, the joint combination of these tuning properties in each STRF characterizes the spectrotemporal processing at each site. To jointly characterize spectrotemporal processing for each electrode across regions, we transformed the STRFs into their modulation transfer functions (MTFs)[23,27,28]. MTFs are the modulation domain representation of spectrotemporal processing characterized by STRFs[28,29]. They summarize neural processing in terms of spectral and temporal modulation tuning and are frequently used to characterize processing within the auditory system. As shown in Fig. 3E, the ensemble (mean) modulation transfer function across frontal lobe areas and STG is similar. To quantify this, the mean temporal modulation tuning and spectral modulation tuning distributions derived from the MTFs were calculated (Fig. 3F), and no significant differences in temporal or spectral modulation tuning were observed across frontal lobe areas and STG (*p* > 0.05, cosine similarity permutation test of mean spectral and temporal modulation tuning vectors[26]). Together, these results indicate that the short-latency responses in frontal lobe areas and STG encode the same spectrotemporal representations of speech, consistent with parallel spectrotemporal speech representations in the temporal and frontal lobes.

While short-onset (<200 ms) electrodes in the frontal lobe are well-predicted by spectrotemporal speech representations, it is possible that they reflect higher-order linguistic features that are correlated with the speech spectrogram. To test this alternative, we compared encoding performance for STRFs to semantic-based encoding models generated with word vector representations of speech derived from the FASTTEXT data set[30]. We expected that both spectrotemporal and semantic encoding models would perform well in each region, however, they would differ from each other as a function of short or long onset latency. Specifically, we hypothesized that the STRF would be a better model for electrodes with short-onset latencies since these populations reflect direct inputs from low-level areas such as the medial geniculate body or primary auditory cortex.

Confirming our hypothesis, we found that electrodes with short-onset latencies (<200 ms) in IFG, mPreCG, and vSMC show significantly higher STRF model predictions than semantic model predictions, consistent with dominant spectrotemporal representations at those sites (Fig. 4A, *p* < 0.01, Wilcoxon rank-sum test). At longer latencies (>200 ms), spectrotemporal and semantic encoding models were not significantly different, further supporting the notion that these short latency electrodes reflect spectrotemporal representations. Overall, these data demonstrate that neural populations in the frontal lobe with short-onset responses encode spectrotemporal representations of sound similar to populations in more low-level auditory areas and STG.

## White matter tractography and functional connectivity show connections from thalamus and primary auditory cortex to frontal lobe areas

To establish whether the short-latency responses with STG-like spectrotemporal encoding observed in frontal lobe cortex reflects a direct anatomical pathway from lower-level auditory areas, we used diffusion tensor imaging (DTI) tractography. Based on the known connectivity patterns with STG, we hypothesized that the medial geniculate body (MGB) within the thalamus and the primary auditory cortex within Heschl's gyrus are key areas that might have parallel projections to the frontal lobe cortex.

Using data from 842 individuals from the Human Connectome project[31], we calculated white matter tractography in each hemisphere with the MGB and Heschl's gyrus as seed regions. To localize the MGB, we used MNI coordinates in conjunction with the terminal point of white matter tracts from the inferior colliculus to the thalamus (Fig. 5A). We restricted the suprasylvian target region of interest to the total area spanned by the lateral frontal and parietal cortex. We identified tracts from MGB that projected to IFG, mPreCG, and right vSMC (Fig. 5A). We also identified tracts from Heschl's gyrus that projected to vSMC and left IFG.

To test whether identified tracts relate to functional measures, we also calculated resting-state functional connectivity between the medial geniculate body or Heschl's gyrus to all of the cortex using fMRI. For the MGB seed region, there was significant functional connectivity with IFG, mPreCG, and vSMC (Fig. 5B, $p < 5 \times 10^{-5}$ after peak-level family-wise error (FWE) correction for multiple comparisons). For the Heschl's gyrus seed region, there was significant functional connectivity to mPreCG and vSMC (Fig. 5C, $p < 5 \times 10^{-5}$ after peak-level family-wise error (FWE) correction for multiple comparisons).

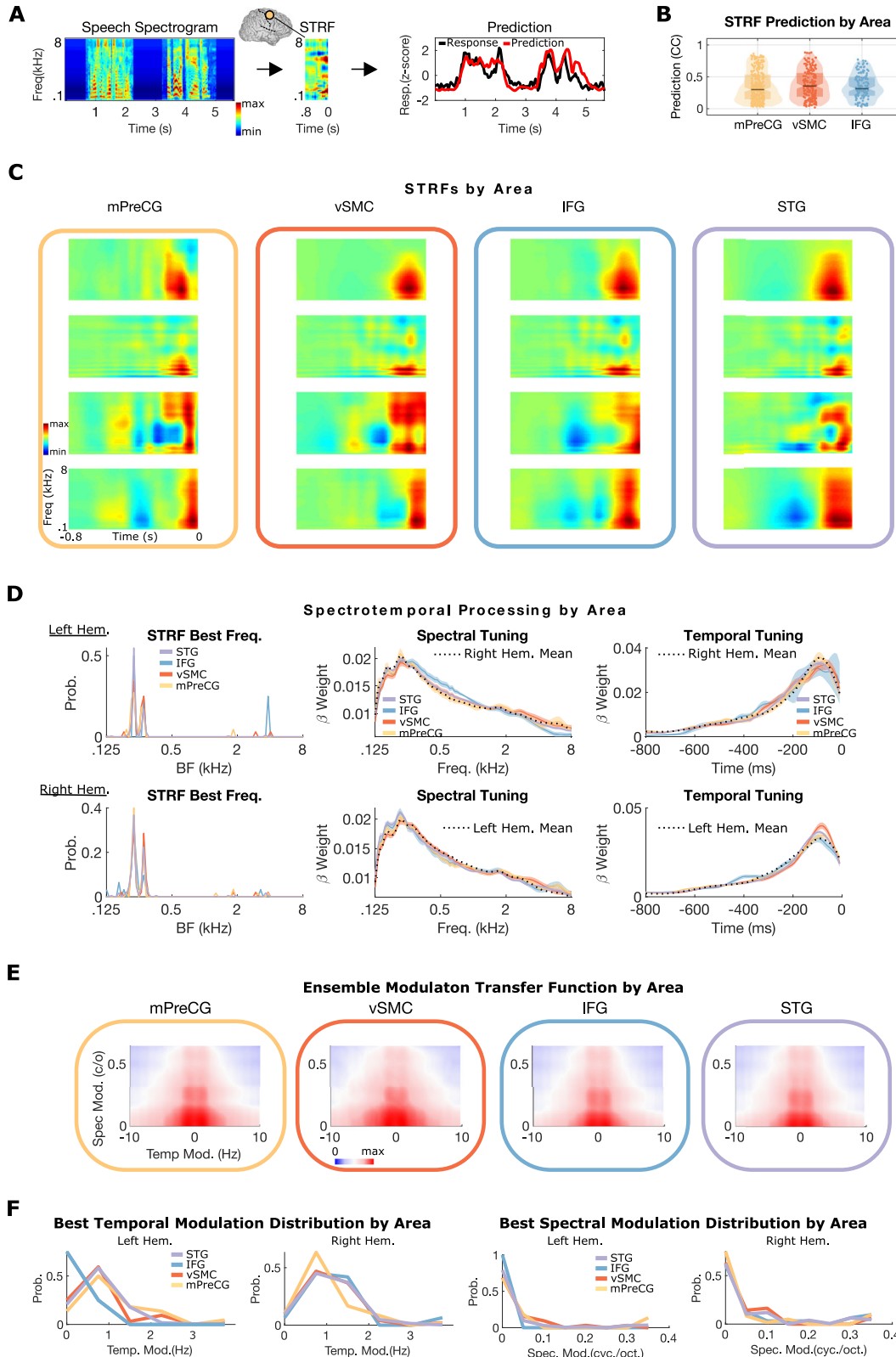

Finally, although limited to the supplementary material due to the scarcity of data (Supplementary Fig. 2), we show single pulse stimulation in the primary auditory cortex with cortico-cortical evoked potentials in these frontal lobe areas consistent with the DTI results above. Together, white matter tractography, resting-state functional connectivity, and single-pulse stimulation demonstrate that short-latency evoked activity that encodes speech-relevant spectrotemporal content in frontal lobe cortex is consistent with a parallel auditory pathway that functions alongside the classical hierarchical speech network mediated by STG.

## Discussion

We examined the nature of short-latency, speech-evoked activity in frontal lobe areas that are not typically associated with auditory

**Fig. 3 | Sites with short-onset latencies in frontal cortex encode the same spectrotemporal information as STG. A** Example middle precentral gyrus STRF with the predicted and actual response to a single sentence. Predicted responses are obtained by convolving the spectrogram with the STRF and are proportional to the similarity between the stimulus's spectrotemporal content and the STRF structure. **B** STRF prediction value distributions for all short-latency electrodes with a significant response to speech in mPreCG, VSMC, and IFG (suprasylvian cortex global mean Pearson correlation coefficient 0.39 ± 0.24; tested on held-out data). STRFs, which are predictive of neural responses, characterize spectro-temporal encoding at each site and are consistent with the presence of spectro-temporal representations in these areas. All box plots show the median, 25th and 75th percentiles. Notches approximate the 95% confidence interval of the median. n site prediction values = 46 IFG, 115 mPreCG, 110 vSMC, and 389 for STG. **C** Examples showing the high degree of STRF similarity between frontal lobe areas and STG.

**D** There is no significant difference in short-onset (<200 ms) STRF tuning para-meters across frontal lobe sites and STG, including: best frequency tuning (p > 0.05 Kruskal–Wallis test), spectral, or temporal tuning (p > 0.05 cosine similarity per-mutation test of mean spectral and temporal tuning vectors). Spectral and tem-poral tuning plots show the mean ± s.e.m for each area in each hemisphere; the total contralateral hemisphere mean tuning is plotted as a dotted line for reference. **E** Ensemble modulation transfer functions by area. The average modulation tuning for each area has a high degree of similarity. **F** There is no significant difference in temporal or spectral modulation tuning between frontal lobe areas and STG (p > 0.05, cosine similarity permutation test of mean spectral and temporal mod-ulation tuning vectors[26]). BF best frequency; CC Pearson correlation coefficient; c/o cycles per octave; Hem. hemisphere; IFG inferior frontal gyrus; mPreCG middle precentral gyrus; Spec. Mod. spectral modulation; STRF spectrotemporal receptive field; Temp. Mod. Temporal Modulation; vSMC ventral sensorimotor cortex.

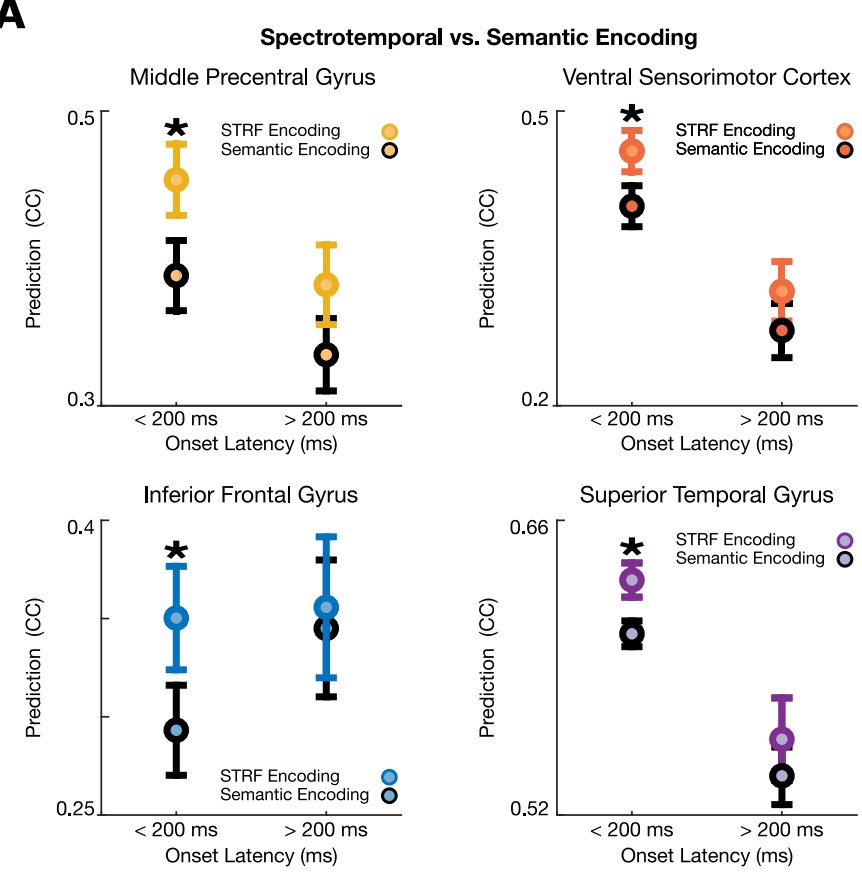

**A**

**Spectrotemporal vs. Semantic Encoding**

**Fig. 4 | Spectrotemporal encoding models are better than semantic encoding models for short-onset latency sites in the frontal lobe and STG. A** Short-onset latency sites in each area encode low-level spectrotemporal information. Sites with onset latencies less than 200 ms show significantly higher STRF model predictions than semantic model predictions consistent with dominant spectrotemporal over semantic coding at these short-onset latency sites. This is consistent with short-onset latency sites encoding spectrotemporal speech information in parallel

(p < 0.001, Wilcoxon rank-sum test, two-sided). Longer onset latency sites in the same areas show no difference between spectrotemporal or semantic encoding models. All data are presented as mean values ± SEM. n site CC prediction values for early (<200 ms)/late (>200 ms) onset latencies = 46/31 IFG, 115/68 mPreCG, 110/43 vSMC, 389/72 STG. CC Pearson correlation coefficient; STRF spectrotemporal receptive field.

processing. Using high-density ECoG, we found that a subset of neural populations in the frontal lobe cortex exhibited response latencies that were synchronous with or preceded the earliest response latencies in STG. Spectrotemporal representations in these populations were lar-gely indistinguishable from those found in STG, encoding key spectral and temporal modulation rates for speech. Finally, we identified white matter tracts that connect early auditory regions like MGB and Heschl's gyrus with frontal lobe areas and were associated with resting-state functional connectivity. Together, these results indicate the

existence of long-range pathways between low-level areas (medial geniculate body, primary auditory cortex) and neural populations in the frontal cortex, which work in parallel to the more well-established hierarchical pathway via lateral cortex areas.

Sensory systems, including vision and hearing, are known to have hierarchical structure, with serial feedforward connections from low-level areas to mid-level and then to apical areas within the frontal cortex[1,2,32]. In addition to this hierarchical structure, locally, within nearby levels, there are short-range parallel connections[3,4].

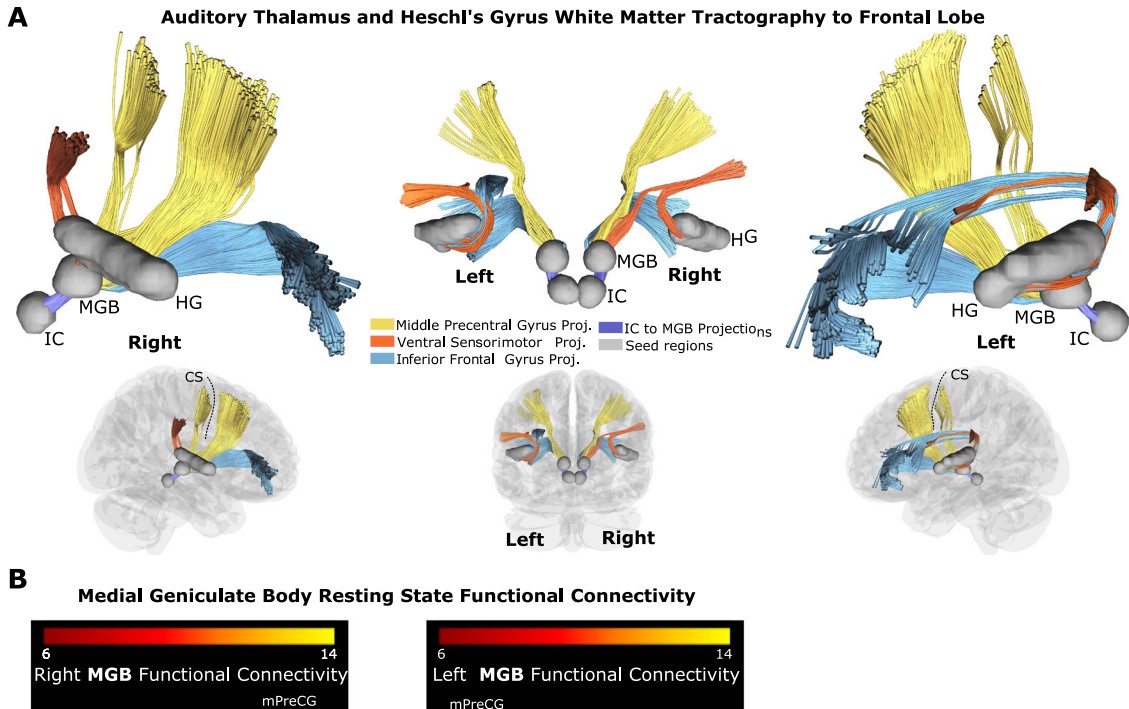

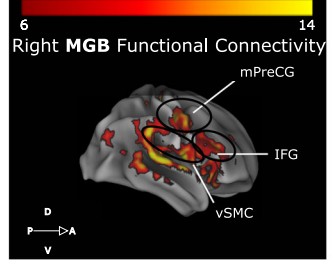

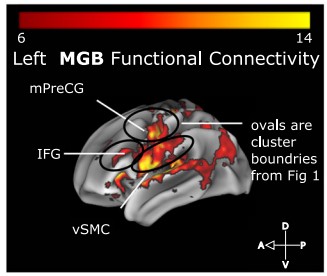

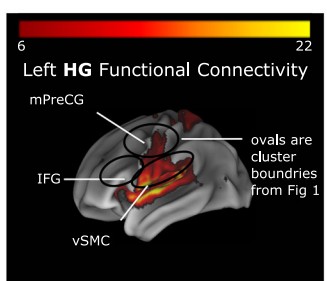

**Fig. 5 | White matter tractography and functional connectivity show connections from thalamus and primary auditory cortex to frontal lobe areas. A** White matter tractography between the medial geniculate body (MGB), Heschl's gyrus (HG), and suprasylvian cortex (IFG, middle precentral gyrus (mPreCG), and ventral sensorimotor cortex (vSMC)). White matter tracts are color-coded by the suprasylvian site of termination. This analysis demonstrates tractography data consistent with direct MGB-to-frontal cortex and HG-to-frontal cortex white matter tracts. **B** Resting-state functional connectivity of the medial geniculate body and **C** Heschl's gyrus showing functional connectivity to frontal lobe areas, middle precentral gyrus, ventral sensorimotor cortex, and inferior frontal gyrus (connectivity maps derived by performing one-sample t-tests, one-sided, with a statistical threshold of $p < 5 \times 10^{-5}$ after peak-level family wise error (FWE) correction for multiple comparisons). Colorbar represents *T*-score. Overall, these data provide additional structural and functional evidence for parallel pathways to frontal lobe areas from low-level areas, such as Heschl's gyrus and the medial geniculate body. CS central sulcus; FC functional connectivity; HG Heschl's gyrus; IFG inferior frontal gyrus; IC inferior colliculus; MGB medial geniculate body; mPreCG middle precentral gyrus; vSMC ventral sensorimotor cortex.

However, the presence of long-range parallel connections in humans from low-level areas, such as the primary auditory cortex or MGB directly to apical areas in frontal cortex have not been described. The present results demonstrate that, in addition to short-range parallel connections, long-range parallel connections exist from the thalamus and primary auditory cortex to areas in the frontal lobe. To what extent these different proposed parallel pathways exist is an important question for further work and could be examined further in participants with depth electrode coverage or tracer studies in primates.

It is important to emphasize the exploratory nature of the tractography results. While focusing on the robust ECoG findings in combination with DTI data, we can present broad interpretation that respects the complexity of the neural architecture involved and produces a model that can be tested with more robust structural analysis of these pathways. DTI-based methods have inherent limitations, including issues of resolution, crossing fibers, and challenges in differentiating close anatomical structures such as the MGB and adjacent thalamic nuclei[33,34]. Although steps within the scope of the study were taken to mitigate these limitations, these findings require further

validation via post-mortem ultra-high-resolution MRI, virtual lesioning[35], tracer injections in primates, and MGB single pulse stimulation in primates[36]. Additionally, future work in which preoperative high-resolution DTI and fMRI recordings in study participants could show that the individual electrode responses are predicted by single-subject connectivity for additional support of the model.

Perhaps a surprising result was how similar spectrotemporal representations encoded by early onset responses are between the frontal lobe and STG. Based on the model of long-range parallel pathways proposed here, frontal neuronal populations and STG would receive the same low-level spectrotemporal representations in MGB or primary auditory cortex to serve as raw snapshot of the of incoming speech signal. This is analogous to work from our group showing primary auditory cortex in Heschl's gyrus, and STG receive rapid parallel inputs that convey spectrotemporal representations of speech[4]. Rapid snapshots of raw spectrotemporal information to frontal lobe areas may help with top-down modulatory functions carried out by the frontal lobe[37–40] or real-time spectrotemporal feedback during speech production[41,42]. Further comparisons between the information encoded in fast frontal lobe responses and longer latency frontal lobe responses (reflecting inputs from canonical hierarchical pathways) is one future direction to investigate the function of these rapid parallel inputs to frontal lobe areas. A second future direction is specialized orthogonal stimuli designed to test which lower-level speech representation is dominant (spectrotemporal versus phonemic versus phonetic feature representations)[21,23,43]. Lastly, due to experimental time constraints, our standard protocol is focused on presenting speech sounds. It is less common to collect data that could speak to questions of whether these early frontal lobe responses are specific to speech. Future work using simple tones, clicks, or white noise bursts could investigate this important question.

It may be surprising to find such long-range connections, particularly as part of a network involved in processing such a complex stimulus. However, three important frontal lobe characteristics suggest a long-range parallel auditory pathway. First, these frontal areas are highly heterogeneous, and sub-populations of neurons in primates appear to be responsive to the acoustic properties of sound, similar to the tuning properties seen in lower-level areas[7]. Second, prior work in primates has suggested the presence of short-onset latencies in the frontal lobe to sound[5,6], and in humans, electrical stimulation of primary auditory cortex elicits short-latency responses in the inferior frontal gyrus[44]. Lastly, in rhesus macaques, non-primary auditory belt regions have tracer-defined anatomic projections to the frontal lobe[45]. The present work shows short-latency neural populations with acoustic representations that are indistinguishable from STG, consistent with long-range parallel transmission of low-level representations of speech directly to the frontal cortex and is further supported by structural evidence of white matter tracts connecting the thalamus and primary auditory cortex to the frontal lobe.

Regarding the potential function of a parallel long-range pathway, an intriguing possibility is that these frontal lobe auditory populations facilitate implementation of processes that are canonically associated with interactions between frontal areas and lower-level auditory areas[37–39,46]. For example, there is substantial data supporting the role of areas like IFG and ventral precentral gyrus in top-down modulation of lower-level areas during speech perception[37–40]. An important aspect of top-down modulation is the necessity for real-time feedback, and consistent with this, experimental evidence has demonstrated the remarkable speed with which it occurs; for example, there is behavioral and neural evidence for nearly instantaneous perceptual warping in phenomena like phoneme restoration[38,47,48]. Whereas most hierarchical models of speech processing in humans assume that the frontal lobe receives inputs from the dorsal and ventral pathways, near instantaneous frontal cortex-mediated perceptual restoration may be facilitated by long-range parallel input to frontal areas that provide a snapshot of the raw incoming sensory data for real-time top-down modulation.

Similarly, a parallel auditory pathway to neural populations throughout sensorimotor speech cortex may play a role in real-time acoustic feedback for articulatory control[41,42]. Neighboring—or possibly even overlapping—neural populations that directly control articulation[49–53] show sensory responses during passive listening[53,54], which could be used in both speech-motor planning and for real-time feedback from the acoustics generated by articulatory output. In either potential role (top-down modulation or real-time acoustic feedback), this long-range parallel pathway gives frontal lobe cortical areas real-time access to the primary spectrotemporal input signal in addition to higher-level representations the frontal lobe receives from other areas, such as the lateral temporal cortex.

We do not disregard the importance of canonical processing pathways, or the fundamental hierarchical nature of the brain[1,2,55]. Indeed, for speech, it is well-established that structures like the arcuate fasciculus support a major pathway between auditory, temporal and frontal cognitive and motor regions[56–58]. Instead, we propose that parallel inputs provide additional computational resources that support the rapid processing required for sounds like speech. Further work is necessary to understand how these pathways work together and whether they have distinct targets within the frontal cortex (perhaps suggested by the distributed and relatively sparse nature of short-latency responses in the present results). However, taken together, the existence of a direct, long-range parallel auditory pathway to the frontal cortex supports the notion that the degree to which neural systems underlying speech perception are largely hierarchical needs to be examined more closely.

## Methods

### Participants and neural recordings

ECoG arrays (interelectrode distance = 4 mm) were placed subdurally in 17 patient volunteers (9 right hemisphere, 8 left hemisphere) undergoing a neurosurgical procedure for the treatment of medication-refractory epilepsy. All participants were native English speakers; all were fluent in English. All participants had normal hearing and no communication deficits. All experimental protocols were approved by the University of California, San Francisco, Institutional Review Board and Committee on Human Research. Each participant gave written informed consent before participating in the study. The location of array placement was determined by clinical criteria alone. Participants were asked to passively listen to 10–40 min of natural speech while ECoG signals were recorded simultaneously. Signals were amplified and sampled at 3052 Hz. After the rejection of electrodes with excessive noise or artifacts, signals were referenced to a common average, and the high-gamma band (70–150 Hz) was extracted as the analytic amplitude of the Hilbert transform[9]. Signals were subsequently downsampled to 100 Hz. For onset latency analysis the high-gamma band was also extracted using Morlet wavelet decomposition and downsampled to 1000 Hz. The resulting signal for each electrode was z-scored based on the mean and standard deviation of activity during the entire block.

### Stimuli

Speech stimuli were delivered binaurally through free-field speakers at approximately 70 dB average sound pressure level. The frequency power spectrum of stimuli spanned 0–8000 Hz. The stimulus set consisted of prerecorded (2–4 s) sentences from the phonetically transcribed TIMIT speech corpus with one-second silent intervals between each sentence presentation[8]. To quantify response characteristics to individual sentences for responsivity and onset latency analysis we analyzed responses to ten unique sentences (each unique sentence repeated ten times). Each participant was also presented an additional 115–489 unique, non-repeated sentences for receptive field

analysis. The total speech corpus included 286 male and 116 female speakers, with 1–3 sentences spoken per speaker, and unique lexical content for each sentence.

## Responsivity and spatial analysis

Ten TIMIT sentences were presented randomly ten times in each participant (100 sentence presentations total). Neural data was aligned by the onset of sound for each sentence. The mean evoked high-gamma to each of the ten sentences was then computed. To test for speech-evoked responses, for each time bin of the evoked high-gamma after sound onset, a Wilcoxon rank-sum test was performed to test for a significant difference from baseline ($p < 0.05$, Bonferroni corrected for the number of sample bins in the sentence). To visualize electrode coordinates in MNI space, we performed nonlinear surface registration using a spherical sulcal-based alignment in Freesurfer, aligning to the cvs_avg35_inMNI152 template[59]. This nonlinear alignment ensures that electrodes on a gyrus in the participant's native space remain on the same gyrus in the atlas space, but does not maintain the geometry of the grid. For spatial analysis, significant responses were projected to the left hemisphere in all participants, given that there were no significant differences in the proportion of responsive sites in each area or amplitude of responses between the left and right hemispheres in the suprasylvian cortex. Spatial clustering was performed using a mixture of Gaussians model with the number of clusters (cluster number = 3) identified by both Bayesian information criterion and Silhouette criterion.

## Analysis

**Onset latency analysis.** To compute the onset of local neuronal spiking, each trial of the repeated TIMIT sentences was aligned by the onset of sound, and the mean evoked high-gamma activity for each sentence was computed. Like prior latency analysis using ECoG in human participants, the high-gamma band, as opposed to the full event-related potential (ERP), was used[60]. High-gamma is used because of its selectivity for neuronal spiking[10–12,61,62]. Additionally, high frequencies that compose the high-gamma band attenuate quickly as a function of distance making them selective for local neuronal activity under each electrode rather than pooling over large areas of cortex that span multiple gyri[63,64]. ERP waveforms from subdural grids also have extremely high waveform shape variability due to the wide pooling of local and volume conducted potentials, making them less tractable for onset latency measurements with a single standardized metric. To compute onset latency, the 500 ms before sentence onset served as the baseline for comparisons. For each 1-ms bin of the evoked high-gamma after sound onset, a Wilcoxon rank-sum test was performed to test for a significant difference from baseline ($p < 0.05$). Like prior ECoG response latency work in the human auditory cortex or primate auditory cortex, response latency was defined as the time in which the mean evoked high-gamma was significantly different from baseline and remained significant for 15 ms[19,20]. The shortest sentence onset latency for each electrode was defined as the speech onset latency for that site.

**Encoding analysis.** STRF and semantic encoding models were fit using normalized reverse correlation[65] with open-source code available at: http://strfpak.berkeley.edu/. Regularization was controlled by fitting a tolerance hyperparameter via cross-validation[66]. STRFs were computed on an estimation set (90% of the total data) and cross-validated on a test set, which was withheld from the estimation process (10% of the data). For STRF fitting, spectrogram representations of speech stimuli were generated using a cochlear model of auditory processing[67]. For the semantic encoding model, word vector representations of the TIMIT speech corpus were derived from the FAS-TTEXT data set[30] by mapping each 10 ms segment of the speech signal to the corresponding word vector representation for that word.

**Modulation tuning.** To characterize modulation tuning, the modulation transfer function (MTF) for each site was computed by taking the magnitude of the two-dimensional Fourier transform ($\mathscr{F}_2\{\bullet\}$) of each STRF:

$$\mathrm{MTF}(\omega_t, \omega_s) = \left| \mathscr{F}_2\{\mathrm{STRF}(t, f)\} \right| \qquad (1)$$

Where $(t, f)$ are time and frequency and $(\omega_t, \omega_s)$ are temporal and spectral modulation, respectively[23].

**Tractography analysis.** For white matter tractography, we applied a deterministic diffusion fiber tractography algorithm[68] using the spin distribution function (SDF) template created by Yeh et al.[69] and publicly available diffusion data (http://brain.labsolver.org) with high angular and high spatial resolution from 842 individuals from the Human Connectome Project[31]. Our tractography analysis used deterministic diffusion tensor imaging (DTI) methods applied to high-resolution data from the Human Connectome Project. Seed regions were placed in the medial geniculate body (MGB) and Heschl's gyrus, with a target region of interest that spanned the lateral frontal and parietal lobe. To mitigate the impact of crossing fibers, we utilized constrained spherical deconvolution (CSD) methods[68], which improve the differentiation of multiple fiber populations within a voxel. Tractography seed and target regions, with exception of the medial geniculate body, were specified using the Human Connectome Project Multi-Modal Parcellation version 1.0 anatomical atlas and the FreeSurfer Destrieux anatomical atlas[70–72]. The medial geniculate body seed region coordinates were determined from previously reported MNI coordinates[73–77] and validated by demonstration of positive tractography from the inferior colliculus to the thalamus. The inferior colliculus is the auditory input nucleus to the medial geniculate body so this is a useful approach to increase confidence of MGB localization and a precise seed region. These inferior colliculus-to-MGB DTI tracts are shown in Fig. 5A (purple tracts). The MGB seed region was further validated by fMRI results (Fig. 5B) that are consistent with what we would expect with an accurate MGB seed region: functional connectivity primarily to planum temporale, STG, and the frontal lobe areas of interest.

**Functional connectivity analysis.** Brain functional images were acquired in a cohort of 50 neurologically intact participants at the Siemens 3-Tesla Prisma scanner located at UCSF. We collected 560 T2*-weighted EPI volumes for each individual with the following parameters: TR/TE = 850/32.8 ms, flip angle = 45°, voxel size = $2.2 \times 2.2 \times 2.2$ mm³, field-of-view = $21 \times 211$ mm², multi-band accelerating factor = 6. Image preprocessing consists of slice-time correction, realignment to the mean functional image, assessment for rotational and translational head motion, and correction for susceptibility-induced distortions. Each participant was assessed for cognitive and language intactness and participants with excessive motor movements during the scan were excluded. We ensure alertness during the scan through continuous real-time monitoring and post-scan participant feedback. Functional images are then normalized to the EPI template in the MNI space with a combination of rigid, affine, and nonlinear warping. After smoothing the images with a 5 mm full width at half maximum (FWHM) Gaussian kernel, CSF and white matter tissue probability maps were then used to compute the mean time series used as regressors. Functional data were then bandpass filtered (0.008 Hz $<f<$ 0.15 Hz), and the nuisance variables were regressed out from the data, which included the six motion parameters, the first derivative and quadratic terms, as well as CSF and white matter time series. Seed ROIs were located bilaterally in the MBG (MNI coordinates: left x = −9, y = −23, z = −1; right x = 8, y = −23, z = −1) and in Heschel's gyrus. Single-subject correlation maps were generated by calculating the $r$–Pearson correlation coefficient between the average BOLD signal time course from the seed ROIs and the time course from all other voxels of the brain. Finally, correlation maps were converted to

$z$-scores, and group-level connectivity maps were calculated for each seed with a statistical threshold at $p < 5 \times 10^{-5}$ after peak-level family-wise error (FWE) correction for multiple comparisons.

## Reporting summary

Further information on research design is available in the Nature Portfolio Reporting Summary linked to this article.

## Data availability

The human patient data relevant to this study are accessible under restricted access according to our IRB protocol. The de-identified patient data for which patients have consented to public release will be made available from the corresponding author upon request. Source data are provided with this paper. The Dryad link to the Source data is specified in the GitHub source code README file.

## Code availability

The analysis and data visualization code will be made available upon request. Source code are provided with this paper: https://github.com/ChangLabUcsf/Hullett_2025_Source_Code

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

## Acknowledgements

We would like to thank participants, members of the Chang lab, and the EEG technologists at the University of California, San Francisco for their time and effort in acquiring electrophysiological data. Research reported in this publication was supported by the National Center for Advancing Translational Sciences of the NIH under Award Number (5TL1TR001871-05, P.W.H.). This work was supported by grants from the National Institute of Deafness and Other Communication Disorders (R01-DC012379, E.F.C., K24 DC015544, M.L.M), the National Institute of Neurological Disorders and Stroke (U01-NS117765, E.F.C., RF1 NS050915 M.L.G.), and the National Institute on Aging (P01 AG019724, M.L.G). This research was also supported by Bill and Susan Oberndorf, the Joan and

Sandy Weill Foundation (E.F.C.), and the William K. Bowes Foundation (E.F.C.).

## Author contributions
Conceptualization, P.W.H., M.K.L., and E.F.C.; investigation, P.W.H.; formal analysis, P.W.H. and M.L.M.; data curation, P.W.H. and M.L.M., writing–original draft, P.W.H.; writing–review and editing P.W.H., M.K.L., M.L.G., M.L.M., and E.F.C.; supervision M.K.L. and E.F.C.; resources, M.L.G. and E.F.C.

## Competing interests
The authors declare no competing interests.
