## [Transparent Peer Review File · Nature Communications]

Parallel encoding of speech in human frontal and temporal lobes

Corresponding Author: Professor Edward Chang

Version 0:

Reviewer comments:

Reviewer #1

(Remarks to the Author)

This is an extremely interesting study of ECoG responses to natural speech across multiple cortical regions, showing equally fast responses in inferior frontal and prefrontal cortex compared to those from auditory regions in superior temporal gyrus. Most novel is the finding of relatively early responses in multiple frontal regions that appear to retain spectrotemporal information, as suggested by the STRF fitting. This is an important result for our understanding of the wider scope of auditory processing across audiomotor regions in cortex (and for cortical organization more generally) and as the authors rightly state, provides vital empirical data for weighting different theoretical accounts of the neurobiology of language.

One additional suggestion for the ECoG data: I suspect that some readers may wonder how much of the spectrotemporal information is encoded at a phoneme or syllable level as compared to a more auditory representation. It would be useful to have a decoding analysis similar to those run by this group, pitting STRF/MTF models versus the more abstract phonemic encoding.

Before addressing the MRI-based sections, I want to emphasize that the ECoG data and analysis stand on their own as a very significant addition to a number of fields and will definitely stimulate and inform discussion in the next years.

While it is terrific to join up different methods to provide a more comprehensive explanatory account for the findings, I think the functional connectivity and particularly HCP-based DTI connectivity analyses have the potential to cloud the otherwise compelling results. As a number of groups have shown (e.g., Thomas, Ye, et al., 2014, PNAS), even the very best diffusion-based tractography estimates (from methods almost impossible to obtain with in-vivo humans) will reveal anatomical connections that do not exist, as well as miss those that are well-documented. Compared to the resolution likely needed to estimate the complex auditory projections from MGB (and most importantly differential connections from tonotopically mapped ventral MGB compared to other subnuclei), the HCP data and the ROI definition do not really approach this level. (See discussions of limitations for much higher resolution subcortical auditory data in Sitek, Gulban, et al (2019)). The claims of novel connections between MGB and frontal regions are very strong, particularly given the geometry of the ROIs compared to the actual anatomy, potential overlap with adjacent large fiber tracts, and the always difficult crossing fiber problem. The definition of the functional MGB ROIs (hard!) is also likely to affect the functional connectivity results. For instance, the connectivity map for right MGB shows that somatosensory and visual regions in the temporoparietal junction appear to be showing connectivity, along with the crown of the middle temporal gyrus. I do not want to say these are completely implausible targets, but I think a much more comprehensive analysis would be needed to

I am sure that the authors have considered this already, but if there are stimulation data available from the frontal sites with recording from auditory regions, these would help to inform the results further.

In sum, I want to emphasize once again that the ECoG results are very important in their own right and will justifiably stimulate a lot of discussion and further studies. It is obviously just one opinion, but I think that leaving the fMRI and DTI results to a different and more speculative paper would be helpful for presenting both datasets.

(Remarks on code availability)

Reviewer #2

(Remarks to the Author)

This study examined speech perception pathways, testing for the existence of a parallel pathway from auditory regions of the thalamus to frontal regions using high-density ECoG, high-resolution DTI, and fMRI functional connectivity. This multi-method approach provides indirect and moderate support for the presence of this alternative pathway. Specifically, the authors report that frontal neuronal populations showed similar latency onsets and similar spectrotemporal responses to those in the STG. The findings were supported by white matter tractography and functional connectivity consistent with direct connections from the auditory nucleus of the thalamus to the frontal lobe, separate from those originating in auditory cortex. While these results are consistent with a model in which the MGB sends direct inputs to frontal regions in parallel with auditory cortex, a number of alternative possibilities need to be examined to increase support for this model.

Major Comments

It is unclear in the main text whether onset latencies are calculated from high gamma power or the unfiltered voltage (raw LFP). The methods state that the mean evoked potential was used for calculation of onset latency. Can the authors please confirm that this was done on the raw LFP and add this information to the main text? This metric is in question because the Results section begins with the statement that data were filtered to the 70-150 Hz (high gamma power) range and it appears high gamma was used for all analyses. If high gamma power was indeed used for latency onset analyses, then ERPs should be examined as well (see next comment).

These data need to be considered in the context of past animal and iEEG studies that have shown early (<20 ms) activation of early auditory cortex. The evidence provided in this submission argues that some frontal lobe responses were earlier than those seen in the STG at the single subject level. However, this is weak evidence dependent on a null effect; the absence of early STG responses in an individual is likely due to the specific cortical coverage in that person, not evidence that the frontal lobe responded more quickly than any auditory region because not all auditory regions were sampled.

The use of subdural recordings ignores the contribution of processing from more medial cortical structures that process sounds rapidly. Indeed it is possible that early auditory cortical structures transmit information in parallel to the lateral STG and to frontal regions. Given that monosynaptic transmission takes <10 ms, and multiple synapses along a direct route 10-30 ms (e.g., Calvin et al., 1976) activation in the frontal lobe would need to be very fast relative to activation in auditory cortex (I'd argue a frontal ERP would need to be generated within 20 ms following sound onset to a simple tone or click).

More information on the tractography procedure is needed and alternatives would be beneficial. Was a second-level refinement of the tractography pathways applied to ensure that the observed results were not due to crossing fibers or fascicles that are better explained by terminations to neighboring regions of the thalamus? Specifically, virtual lesioning approaches such as those from Pestilli et al., 2014 Nature Methods can help examine the amount of variance a specific pathway explains relative to whole brain diffusion data (i.e., whether the MGB-Frontal tract explains unique variance). This is an important consideration given the proximity of other non-auditory nuclei in the thalamus that are known to have projections to frontal regions.

A broader discussion is needed for how, based on this model, frontal neuronal populations would acquire similar spectrotemporal responses to those in the STG while being transmitted from the MGB, compared to the alternative possibility that these patterns originate in the STG.

Minor Comments

fMRI functional connectivity was estimated from a different sample of individuals from the large sample of HCP participants used for DTI analyses. Can the authors clarify why they did not use fMRI resting state activity from the same HCP subjects? This seems like a missed opportunity that would allow for a more fine grained analysis of connectivity at the individual subject level and to avoid cross-subject variability in path terminations.

Are DTI and fMRI resting state available for any of the ECoG subjects? Showing that the individual electrode responses are predicted by single-subject connectivity patterns would add support for this model.

Do the authors have causal evidence to support these data? E.g., any patients with stimulation within the thalamus who have recording sites in these frontal regions? Or patients with auditory cortex damage who show auditory responses at these frontal sites are preserved?

The authors show convincingly that these frontal regions demonstrate similar spectrotemporal sensitivity as regions in the lateral STG. These results would benefit by an analysis of responses to simple tones, clicks, or white noise bursts (if any of those data were collected for these patients). This would show whether the frontal responses are selective to speech and enable more reliable estimates of timing in the two regions.

(Remarks on code availability)

Reviewer #3

(Remarks to the Author)

1. What are the noteworthy results?

- A) Current theories largely support the principle that frontal cortical responses to speech arise primarily from sequential hierarchical speech feature encoding.
- B) This study provides strong evidence supporting an innovative hypothesis that short-latency (<200 ms) frontal cortical responses in humans are driven by parallel ascending auditory pathways from the auditory thalamus and primary auditory cortex.
- C) The dual parallel and hierarchical pathway organization for speech and other sound processing has strong potential to radically change our computational models, simulations, and understanding of how speech and sensory inputs modulate high-level frontal cortical processes.

2. This study is noteworthy for the following reasons.

- A) This study provides strong evidence supporting the hypothesis that short-latency (<200 ms) frontal cortical responses in humans are driven by parallel ascending auditory pathways from the auditory thalamus and primary auditory cortex.
- B) ECoG provides high spatial and temporal resolution metrics of frontal, superior temporal gyrus (STG), and primary auditory cortical (PAC) neural activity.
- C) They find that "lower level" speech-specialized superior temporal gyrus (STG) and three separate higher level prefrontal cortical areas (mPreCG, vSMC and IFG) respond to speech onset with similarly short latencies (<200 ms). This supports prior work demonstrating short latency auditory evoked ECoG responses in prefrontal cortices in primates (Romanski & Hwang Neurosc 2012; Marrouch et al., Annals of Mathematics and Artificial Intelligence 2020).
- D) Importantly, it supports the general principle that prefrontal cortical areas receive parallel ascending thalamic and primary auditory cortical input in addition to sequential auditory cortical pathway inputs.
- E) They find that short-latency speech-driven Spectral-Temporal Response Fields (STRFs) are similar within the speech-specialized superior temporal gyrus (STG) as well as three prefrontal cortical areas (mPreCG, vSMC, and IFG).
- F) Importantly, the speech-driven STRF responses in prefrontal and STG areas both have neural responses that are predictive of the spectral and temporal modulations in speech sounds.
- G) They find that STRF spectrotemporal encoding models predict spectral and temporal features in speech better than semantic encoding models, for short-onset latency sites in the frontal lobe and STG.
- H) Finally, white matter tractography and functional magnetic resonance imaging (fMRI) metrics provide evidence of parallel anatomic and functional neural pathway connections between the auditory thalamus and the three frontal cortical areas (mPreCG, vSMC, and IFG) in addition to primary auditory cortex on Heschl's gyrus.

3. This work will be significant for general auditory and speech processing fields and potentially more broadly will help define novel principles for how ascending sensory responses modulate top-down frontal cortical processes and executive control.

4. How does it compare to the established literature?

- A) These ECoG recordings provide an unparalleled, high degree of spatial and temporal resolution that is not possible with standard electroencephalogram (EEG) and magnetoencephalography (MEG) measures.
- B) Coupling this with the tractography and fMRI analysis provides strong support for their hypothesis of parallel short-latency ascending pathway inputs to primary auditory cortices as well as prefrontal cortices.

5. If the work is not original, please provide relevant references.

- A) This work is original in that it clearly demonstrates similar speech-evoked short-latency responses in frontal and STG areas with high spatial and temporal resolution afforded by intracranial ECoG.
- B) Though they cite Romanski & Hwang Neurosc 2012 for demonstrating short latency auditory evoked responses in prefrontal cortices in primates, they could also cite (Marrouch et al., Annals of Mathematics and Artificial Intelligence 2020) who demonstrate similarly short latency auditory evoked ECoG responses in primates (Fig. 4).

6. Does the work support the conclusions and claims, or is additional evidence needed?

- A) This work provides strong support for their general conclusions—no additions are needed.
- B) In the Discussion, they may want to briefly explain that future studies could compare speech-related features encoded in the short-latency versus longer latency responding sub-columns within prefrontal and STG areas which they touch upon.

7. Are there any flaws in the data analysis, interpretation and conclusions?

- A) No major flaws are evident.

8. Do these prohibit publication or require revision?

This manuscript reads well for a general and specialized audiences –I think it is ready to go!

9. Is the methodology sound?

- A) Yes, the methods are sound, strong and relevant for addressing this novel hypothesis.

10. Does the work meet the expected standards in your field?

- A) This meets high standards in the field. The same collaborators are experts in this area with publications demonstrating

regional topography and specialization for speech-processing within STG using ECoG, STRFs, and MTF analytic techniques (Hullett, P. W., Hamilton, L. S., Mesgarani, N., Schreiner, C. E. & Chang, E. F. Human superior temporal gyrus organization of spectrotemporal modulation tuning derived from speech stimuli. J Neurosci <https://pubmed.ncbi.nlm.nih.gov/26865624/>)

11. Is there enough detail provided in the methods for the work to be reproduced?

A) Yes, there is ample detail in the Methods to reproduce this work.

REVIEWER: Heather L. Read, University of Connecticut, Psychological Sciences & Biomedical Engineering.

(Remarks on code availability)

I logged into github and went to the github URL: <https://github.com/ChangLabUcsf/Hullett2024>

However, it says, "This repository is empty."

Version 1:

Reviewer comments:

Reviewer #1

(Remarks to the Author)

Thanks to the authors for the responses, and revisions to the manuscript, the additional details and discussion really hit the mark, and addressed my concerns. It has been a pleasure to learn about this work, and I look forward to sharing it with colleagues when it's published.

(Remarks on code availability)

Reviewer #2

(Remarks to the Author)

The authors have addressed my previous comments. This study uses a unique and highly valuable set of data to provide evidence for the fast transmission of spectrotemporal speech information to the frontal lobe, potentially through parallel cortical and thalamic pathways. It will likely have a strong impact on the field.

(Remarks on code availability)

Reviewer #3

(Remarks to the Author)

The updates look excellent. Seems they addressed all three Reviewers concerns.

(Remarks on code availability)

The updated manuscript addresses all the primary concerns raised by the reviewers.

This study has the potential to impact our general theories and computational models for how the brain processes speech and to impact the development of novel AI and Machine Learning models to perform similar processes. Moreover, this has the potential to expand approaches to treatments for various disorders that impact speech.

Minor Edit: Lines 287-289: "A second future direction is specialized orthogonal stimuli designed to test which lower-level speech representations is dominant..." change "speech representations is dominant.." to "speech representation is dominant".

Reviewer 1 Comments and Responses

This is an extremely interesting study of ECoG responses to natural speech across multiple cortical regions, showing equally fast responses in inferior frontal and prefrontal cortex compared to those from auditory regions in superior temporal gyrus. Most novel is the finding of relatively early responses in multiple frontal regions that appear to retain spectrotemporal information, as suggested by the STRF fitting. This is an important result for our understanding of the wider scope of auditory processing across audiomotor regions in cortex (and for cortical organization more generally) and as the authors rightly state, provides vital empirical data for weighting different theoretical accounts of the neurobiology of language.

We appreciate the Reviewer's extremely positive assessment of the paper, and their recognition that this work provides important contributions to theoretical accounts of the neurobiology of language.

Comment1.1:

“One additional suggestion for the ECoG data: I suspect that some readers may wonder how much of the spectrotemporal information is encoded at a phoneme or syllable level as compared to a more auditory representation. It would be useful to have a decoding analysis similar to those run by this group, pitting STRF/MTF models versus the more abstract phonemic encoding.

Response1.1:

We thank the Reviewer for suggesting this point, as it raises an interesting question about linguistic encoding in these frontal regions. While the Reviewer is correct that phonemic (and acoustic-phonetic) encoding has been characterized in areas like STG, it is important to note that we have not found it fruitful to directly compare STRF/MTF models to these more abstract stimulus encodings. There are two main reasons for this: (1) the feature spaces are too highly correlated, and (2) combined with the stimulus correlation, the vastly different numbers of features in spectrotemporal (~4000) vs acoustic-phonetic (~600) makes model comparison challenging.

Regarding point (1), the stimulus feature spaces are inherently correlated because they describe the same signal. In particular, manner of articulation features (e.g., plosive, fricative, nasal, etc) have clear and direct acoustic correlates in the spectrogram. While place of articulation (e.g., coronal, dorsal, velar, etc) are less well-defined acoustically, they are still present. Ultimately, this means that in an analysis that compares the unique variance (e.g., ΔR^2) between spectrotemporal and acoustic-phonetic features has little opportunity to find variance in one feature space that isn't also attributable in the other.

Regarding point (2), the different numbers of features (the numbers above are computed based on the count of features x time lags) presents further problems for model comparison. Whereas measures like unique variance, BIC, or AIC are designed to account for different numbers of features (in the latter cases by explicitly penalizing the size of the feature space), this only works in practice when the feature spaces being compared are sufficiently uncorrelated.

We have attempted to perform these comparisons on similar datasets in the past, but have not
found a satisfactory approach. For this reason, we did not include such a comparison here.
Furthermore, given the relationship between spectrotemporal/modulation and phonemic tuning,
we believe it is possible to make broad claims about the latter based on the results we present.

Therefore, to address this point, we have added qualitative descriptors of tuning to acoustic-
phonetic/phonemic features for example electrodes (*see lines 166-169*). We have also added a
statement to the Discussion about how future work could use controlled stimuli to more
quantitatively evaluate the encoding of abstract phonological features in frontal areas, which
would help further elucidate these areas' role in speech perception (*see lines 287-290*)

**Comment1.2:**

“Before addressing the MRI-based sections, I want to emphasize that the ECoG data and analysis
stand on their own as a very significant addition to a number of fields and will definitely
stimulate and inform discussion in the next years. While it is terrific to join up different methods
to provide a more comprehensive explanatory account for the findings, I think the functional
connectivity and particularly HCP-based DTI connectivity analyses have the potential to cloud
the otherwise compelling results. As a number of groups have shown (e.g., Thomas, Ye, et al.,
2014, PNAS), even the very best diffusion-based tractography estimates (from methods almost
impossible to obtain with in-vivo humans) will reveal anatomical connections that do not exist,
as well as miss those that are well-documented. Compared to the resolution likely needed to
estimate the complex auditory projections from MGB (and most importantly differential
connections from tonotopically mapped ventral MGB compared to other subnuclei), the HCP
data and the ROI definition do not really approach this level. (See discussions of limitations for
much higher resolution subcortical auditory data in Sitek, Gulban, et al (2019)). The claims of
novel connections between MGB and frontal regions are very strong, particularly given the
geometry of the ROIs compared to the actual anatomy, potential overlap with adjacent large fiber
tracts, and the always difficult crossing fiber problem. The definition of the functional MGB
ROIs (hard!) is also likely to affect the functional connectivity results. For instance, the
connectivity map for right MGB shows that somatosensory and visual regions in the
temporoparietal junction appear to be showing connectivity, along with the crown of the middle
temporal gyrus. I do not want to say these are completely implausible targets, but I think a much
more comprehensive analysis would be needed to” [... the remainder of this comment was not
present in the received reviews.]

**Response1.2:**

We thank the Reviewer for their thoughtful comments and for highlighting the strength of the
ECoG findings. We appreciate the concern regarding the potential limitations of diffusion
tractography, particularly in the context of estimating complex auditory projections. Our
intention was not to assert that our findings represent the discovery of entirely novel tracts but
rather to provide additional evidence for the presence of potential long-range parallel pathways
that complement existing hierarchical models of speech processing.

In general, we want to emphasize that we agree with the Reviewer that the primary strength of
the paper lies in the novel ECoG results. That said, given the novelty of these findings, we
believe it is important to back them up with as much additional evidence as possible. While we

want to avoid confirmation bias, finding concordant results with additional data modalities like
DTI – taking into account the important caveats raised above and discussed in detail below –
provides additional confidence for our interpretation of the key results in the paper. Therefore,
the changes we have made to the manuscript are meant to situate these data in the proper context,
and to provide the important caveats that apply. We believe this approach provides readers with
the information necessary to understand and interpret the DTI data and how it is used here to
support our claims.

To address the specific concerns:

- 1. **Clarification of Claims:** We did not intend to imply that we have identified previously
unknown anatomical connections. Instead, we aimed to highlight that our diffusion-based
tractography findings suggest pathways that might align with earlier latency responses
observed in ECoG, thus supporting the possibility of parallel projections from the MGB
to frontal areas. We have revised the manuscript to emphasize that these results should be
interpreted as supporting the ECoG results and are suggestive of such connections rather
than definitive evidence of novel anatomical pathways.
- 2. **Acknowledging Tractography Limitations:** We fully acknowledge the limitations of
DTI-based methods, including issues of resolution, crossing fibers, and the inherent
challenges in differentiating close anatomical structures such as the MGB and adjacent
subcortical areas. As suggested, we have referenced studies (e.g., Thomas, Ye, et al.,
2014; Sitek, Gulban, et al., 2019) that discuss these limitations and have added text to the
Discussion to clarify that while tractography suggests certain pathways, these findings
require further validation through higher-resolution methods.
- 3. **Functional and Structural Evidence Integration:** We believe that the integration of
ECoG findings with structural connectivity analyses offers a broader perspective on the
organization of auditory projections, even with the limitations of the latter. Our goal was
to combine the strengths of high-temporal resolution ECoG with structural insights from
tractography to provide a more comprehensive view of the network architecture. We have
revised the text to ensure that the ECoG data is emphasized as the primary contribution,
with the DTI and functional connectivity analyses presented as supporting evidence
rather than as the central focus.

*The revised text can be found on lines 264-275 of the manuscript.*

We have also removed the second to last paragraph of the discussion (below) so as to not
overstate the findings.

“ Finally, we demonstrated that short-latency, parallel auditory responses in frontal lobe cortex
are supported by white matter tracts from the medial geniculate body in the thalamus and Heschl’s
gyrus. To our knowledge, these white matter tracts have not been identified previously. This is
likely because the most robust analysis requires a specific hypothesis about seed regions and
targets, and the specific pairing of areas like MGB and frontal cortex may have yet to be
considered. However, it is important to note that for DTI tractography to identify a robust tract, it
must be composed of a relatively large, anisotropic bundle of fibers. Thus, these newly identified
tracts are not minor examples of highly specific connections but rather are substantial, long-range
myelinated fibers.”

**Comment1.3:**

“I am sure that the authors have considered this already, but if there are stimulation data
available from the frontal sites with recording from auditory regions, these would help to inform
the results further.”

**Response 1.3:**

We thank the Reviewer for the suggestion. We agree that this is a very powerful method to
investigate effective connectivity in human participants. Unfortunately, it is very difficult data to
obtain. In our dataset, there is no stimulation in frontal sites with recordings in auditory regions.
However, we were able to add one case of the reverse paradigm which adds similar informative
value. In the new supplemental Figure 2, we show significant cortico-cortical evoked potentials
in these frontal lobe areas with single-pulse stimulation of primary auditory cortex along
Heschl’s gyrus. This provides further evidence of direct parallel connections from Heschl’s
gyrus to the frontal lobe.

**Comment1.4:**

“In sum, I want to emphasize once again that the ECoG results are very important in their own
right and will justifiably stimulate a lot of discussion and further studies. It is obviously just one
opinion, but I think that leaving the fMRI and DTI results to a different and more speculative
paper would be helpful for presenting both datasets.”

**Response1.4:**

Again, we thank the Reviewer for their positive assessment and helpful suggestions. We hope
that the additional data and modifications to the text address these concerns as they relate to the
main claims of this manuscript.

**Reviewer 2 Comments and Responses**

This study examined speech perception pathways, testing for the existence of a parallel pathway
from auditory regions of the thalamus to frontal regions using high-density ECoG, high-
resolution DTI, and fMRI functional connectivity. This multi-method approach provides indirect
and moderate support for the presence of this alternative pathway. Specifically, the authors
report that frontal neuronal populations showed similar latency onsets and similar
spectrotemporal responses to those in the STG. The findings were supported by white matter
tractography and functional connectivity consistent with direct connections from the auditory
nucleus of the thalamus to the frontal lobe, separate from those originating in auditory cortex.
While these results are consistent with a model in which the MGB sends direct inputs to frontal
regions in parallel with auditory cortex, a number of alternative possibilities need to be examined
to increase support for this model.

We thank the Reviewer for their positive assessment of the paper, and for bringing up these
specific questions and critiques. Below, we respond to each comment in detail.

**Comment 2.1:**

It is unclear in the main text whether onset latencies are calculated from high gamma power or

the unfiltered voltage (raw LFP). The methods state that the mean evoked potential was used for
calculation of onset latency. Can the authors please confirm that this was done on the raw LFP
and add this information to the main text? This metric is in question because the Results section
begins with the statement that data were filtered to the 70-150 hz (high gamma power) range and
it appears high gamma was used for all analyses. If high gamma power was indeed used for
latency onset analyses, then ERPs should be examined as well (see next comment).

**Response 2.1:**

We appreciate this comment because it raises an important point. First, we apologize for the lack
of clarity. We used the term “evoked potential” to refer to the high-gamma amplitude activity
evoked in response to the stimulus, not the raw voltage potential signal. We have addressed this
by modifying the text on pages 23 and 24.

Second, because the high-gamma band reflects local neuronal spiking¹⁻⁴, it has been used
extensively in prior work, including to investigate onset latencies in the human auditory system⁵.
We use high-gamma activity in this manuscript to compute the onset of local neuronal spiking
under each electrode and decode the information in those responses because we specifically want
to make claims about when neurons in these areas respond to acoustic input.

While we agree with the Reviewer that the raw voltage ERPs contain important information
about neural activity, the less direct relationship with neuronal firing rates, the less local signals
associated with volume conduction)⁶⁻⁸, and the multiphasic nature of the raw evoked potentials
make quantification of onset latencies challenging.

Regarding this third point, below we show examples of the high-gamma and raw voltage evoked
responses to speech from several participants. As is apparent on the right, the multiphasic raw
voltage response makes computing and interpreting onset latencies challenging. Although the
raw voltage has an advantage of not being affected by smearing associated with filtering, we
have addressed this concern in our use of Morelet Wavelets to compute the high-gamma signal
(see Methods).

Response Figure 3. Contrast between high-gamma responses (A) compared to high waveform morphology of ERPs (B) calculated at the same electrodes. Each row is from a different participant. Each response represents the average to 400 sentences.

To further address this point, we have added text to the revised manuscript (*see lines 387-396*)
 explaining our justification for focusing on the high-gamma amplitude, and noting that future
 work could address how these effects manifest in the more complex multiphasic response in the
 raw voltage signal.

**Comment 2.2:**

These data need to be considered in the context of past animal and iEEG studies that have shown
 early (<20 ms) activation of early auditory cortex. The evidence provided in this submission
 argues that some frontal lobe responses were earlier than those seen in the STG at the single
 subject level. However, this is weak evidence dependent on a null effect; the absence of early
 STG responses in an individual is likely due to the specific cortical coverage in that person, not
 evidence that the frontal lobe responded more quickly than any auditory region because not all
 auditory regions were sampled.

**Response 2.2:**

We thank the reviewer for pointing out this important issue. To clarify our primary claim, we
 only argue that frontal areas have similar latencies to lateral secondary auditory cortex (STG),
 not other parts of the auditory hierarchy, which the Reviewer is correct in saying are not sampled
 here.

At the same time, the Reviewer is correct to note that it is important to understand to what extent
 sampling of the STG itself is limited. We show in Supplementary Figure 1 that all participants
 had as complete high-density STG coverage as possible. Indeed this study is only possible

because of the use of specialized high-density grids that also provide broad coverage of both
superior temporal and frontal regions⁹ (4mm interelectrode distance; standard ECoG grids have
10mm interelectrode distance).

Of course, we take the Reviewer's point that ECoG inherently undersamples the cortex, which
does not rule out the possibility that there are neural populations with faster responses than any
frontal populations. Indeed, as shown in Fig. 2, this effect is not categorically true at the ROI
level – not all frontal populations have an early onset latency (though of course the same is true
for STG). Together, we take these points as evidence that the existence of synchronous or earlier
responses within the frontal lobe relative to some STG populations is consistent with the main
claim of a parallel pathway.

*We have added text referencing the supplemental figure and above points on lines 124-129 of the*
*manuscript.*

**Comment 2.3:**

The use of subdural recordings ignores the contribution of processing from more medial cortical
structures that process sounds rapidly. Indeed, it is possible that early auditory cortical structures
transmit information in parallel to the lateral STG and to frontal regions. Given that
monosynaptic transmission takes <10 ms, and multiple synapses along a direct route 10-30 ms
(e.g., Calvin et al., 1976) activation in the frontal lobe would need to be very fast relative to
activation in auditory cortex (I'd argue a frontal ERP would need to be generated within 20 ms
following sound onset to a simple tone or click).

**Response 2.3:**

We thank the Reviewer for highlighting this important point. We recognize that we were not
clear enough in the original manuscript, but our claim of parallel pathways includes the
possibility of them starting in these medial cortical structures (Heschl's gyrus / primary auditory
cortex (PAC)) or the medial geniculate body, or both. The presence of DTI tracts from Heschl's
gyrus to the frontal lobe (Figure 5) suggest this as a possibility and we have included this
possibility in the hypothesis summary ("Proposed Long-range Parallel Pathway(s)") in Figure
1A.

Regarding the specific comment about synaptic transmission latencies, we agree that it is
difficult to establish precisely how fast we expect latencies to be, both because we lack evidence
here for the number of synapses in each pathway, and because we are dealing with population-
level activity rather than single neurons. Therefore, we believe that the strongest claim can be
made by directly comparing like-to-like with STG and frontal cortex in relative terms, rather
than in absolute latencies.

To address these concerns, we have changed several instances of text in the manuscript to
indicate primary auditory cortex (medial auditory cortex) and the medial geniculate body are
both candidate areas where long-range parallel projections could arise. We specifically note in
the discussion that the lack of depth electrode coverage in these participants leaves open the
empirical question of to what extent these different proposed parallel pathways exist, and that

this is an important question for further work. We also note that our claims are based on the
relative, not absolute latencies measured in each region using ECoG.

**Comment 2.4:**

More information on the tractography procedure is needed and alternatives would be beneficial.
Was a second-level refinement of the tractography pathways applied to ensure that the observed
results were not due to crossing fibers or fascicles that are better explained by terminations to
neighboring regions of the thalamus? Specifically, virtual lesioning approaches such as those
from Pestilli et al., 2014 Nature Methods can help examine the amount of variance a specific
pathway explains relative to whole brain diffusion data (i.e., whether the MGB-Frontal tract
explains unique variance). This is an important consideration given the proximity of other non-
auditory nuclei in the thalamus that are known to have projections to frontal regions.

**Response 2.4:**

We appreciate this valuable feedback and the suggestions. Below and in the revised manuscript,
we provide additional details on our tractography procedure and discuss the approaches we used
to address crossing fibers and exclude neighboring regions/nuclei in the thalamus that may also
have projections to the frontal lobe.

1. **Details on the Tractography Procedure:** Our tractography analysis used deterministic
diffusion tensor imaging (DTI) methods applied to high-resolution data from the Human
Connectome Project. Seed regions were placed in the medial geniculate body (MGB) and
Heschl's gyrus, with a target region of interest that spanned the lateral frontal and parietal
lobe.

304

2. **To mitigate the impact of crossing fibers**, we utilized constrained spherical
deconvolution (CSD) methods¹⁰, which improve the differentiation of multiple fiber
populations within a voxel.

3. **Second-Level Refinement to exclude neighboring nuclei:** First, we employed rigorous
criteria for seed region definition using MNI coordinates and anatomical landmarks, and
small seed region volume to minimize overlap with adjacent non-auditory nuclei.
Secondly, to ensure precise MGB definition, we validated that DTI tracts from the
inferior colliculus to the MNI based MGB seed region were present. The inferior
colliculus is a primary auditory relay nucleus just ventral to MGB in the auditory
pathway. The inferior colliculus primarily projects to MGB so this is a useful approach to
increase confidence of MGB localization and a precise seed region. These inferior
colliculus-to-MGB DTI tracts are shown in Figure 5A (purple tracts). Lastly, this same
the MGB seed region is used for functional connectivity analysis in Figure 5B and
produces the known and expected functional connectivity pattern in the temporal lobe for
accurate MGB localization. For example there is high functional connectivity in the
temporal lobe (planum temporale and STG). If the MGB seed region included adjacent
structures like the lateral geniculate nucleus (LGN), there would be high functional
connectivity in the occipital cortex which we do not see.

4. **Future Directions:** We have also acknowledged the importance of differentiating
projections from other thalamic nuclei in the manuscript and suggest that future work
could integrate ultra-high-resolution diffusion imaging, advanced probabilistic
tractography methods, virtual lesioning (Pestilli et al., 2014), and non-MRI based
structural analysis (anatomic tracer injections and single pulse stimulation in primates).

These techniques could provide further clarity on the specific contributions of the MGB-
frontal pathway relative to neighboring thalamic regions, as well as validate the
robustness of our observed connections.

We hope that this additional information on our methods provides a satisfactory response to this
comment. Crucially, however, we also emphasize that the DTI data serve as additional
supporting evidence for the ECoG results, and because of the limitations noted here, we do not
make claims based solely on the imaging data. As discussed above (and in response to Reviewer
1), we have clarified in the manuscript the supporting nature of these data, and have noted the
limitations that preclude any over-interpretation of the DTI results on their own.

*The added text is below and can be found in at lines 264-275 and 428-443 in the manuscript.*

Lastly, we have removed the following paragraph from the discussion to avoid overly weighting
the DTI analysis:

“ Finally, we demonstrated that short-latency, parallel auditory responses in frontal lobe cortex
are supported by white matter tracts from the medial geniculate body in the thalamus and
Heschl’s gyrus. To our knowledge, these white matter tracts have not been identified previously.
This is likely because the most robust analysis requires a specific hypothesis about seed regions
and targets, and the specific pairing of areas like MGB and frontal cortex may have yet to be
considered. However, it is important to note that for DTI tractography to identify a robust tract, it
must be composed of a relatively large, anisotropic bundle of fibers. Thus, these newly identified
tracts are not minor examples of highly specific connections but rather are substantial, long-
range myelinated fibers.”

**Comment 2.5:**

A broader discussion is needed for how, based on this model, frontal neuronal populations would
acquire similar spectrotemporal responses to those in the STG while being transmitted from the
MGB, compared to the alternative possibility that these patterns originate in the STG.

**Response 2.5:**

We thank the reviewer for this helpful suggestion. Prior work from our group has shown primary
auditory cortex in Heschl’s gyrus and STG receive parallel inputs that convey spectrotemporal
representations of speech¹¹. The results in the present manuscript are an extension of that parallel
architecture model arguing that these frontal lobe areas are receiving these parallel
spectrotemporal representations as well.

Regarding the specific question of why these frontal populations have similar tuning profiles as
STG and not, say, primary auditory cortex, we now argue more explicitly that it is a result of a
shared input through the parallel pathways. There remain open questions about the precise
functions of this additional STG-like activity, and we provide some possibilities in the
Discussion for readers to consider.

*The added text regarding this can be found in the discussion section on lines 276-284 of the*
*manuscript.*

Minor Comments

**Comment 2.6:**

fMRI functional connectivity was estimated from a different sample of individuals from the
large sample of HCP participants used for DTI analyses. Can the authors clarify why they did not
use fMRI resting state activity from the same HCP subjects? This seems like a missed
opportunity that would allow for a more fine grained analysis of connectivity at the individual
subject level and to avoid cross-subject variability in path terminations.

**Response 2.6:**

We appreciate the reviewer's observation regarding the advantages and disadvantages of using a
UCSF-acquired sample of 50 participants to define normative intrinsic functional connectivity
networks rather than the larger HCP data set. We want to clarify that this additional analysis is
not necessary for the main claim, but serves as additional second-order support. By using our
own in-house data set for functional connectivity we were able to more rigorously ensure the
quality. Most importantly we were able to ensure cognitive and language intactness within each
participant. We also restricted inclusion to participants with minimized head motion for the
duration of the scan, minimal motor movements during the scan, and full alertness during the
scan as assessed by continuous real-time monitoring and post-scan feedback. We acknowledge
that the relatively smaller sample size (50 participants) may prohibit more fine grained analysis,
but for the current study, this not required for the central claim.

We have added the above details to the methods section (lines 451-453).

**Comment 2.7:**

Are DTI and fMRI resting state available for any of the ECoG subjects? Showing that the
individual electrode responses are predicted by single-subject connectivity patterns would add
support for this model.

**Response 2.7:**

We agree that the ability to look at all three data modalities in the same participants would be
even stronger than the across-subject results we currently have. Unfortunately, it is not standard
at our center to collect fMRI and high-resolution DTI from epilepsy surgical candidates.
However, it is fruitful idea and something we are in the process of establishing for future cases.

*We have added text to lines 272 – 275 of the revised manuscript noting this limitation.*

**Comment 2.8:**

Do the authors have causal evidence to support these data? E.g., any patients with stimulation
within the thalamus who have recording sites in these frontal regions? Or patients with auditory
cortex damage who show auditory responses at these frontal sites are preserved?

**Response 2.8:**

We agree that this would be helpful additional evidence for our claims. Regarding thalamic
stimulation, it is not standard to implant electrodes in epilepsy patients in the thalamus, therefore
we do not have the opportunity to perform this specific experiment. However, such work could

be done in non-human primates, which we now specify as a fruitful future investigation in the
discussion.

Regarding the second suggestion, we have also never had the clinical scenario to record frontal
lobe speech response in a participant with auditory cortex damage and are not aware of such data
in another lab. However, we recognize that such a clinical presentation is possible (perhaps less
likely in epilepsy than in brain tumor or stroke), and we believe that should such a case report
become available, it would speak to the claims in the present manuscript.

To the reviewer's general point about causal support, we were able to collect single pulse
stimulation data from primary auditory cortex in a single participant and found significant
responses in these frontal lobe areas (Fig. S2). We note that clinical scenarios where the auditory
cortex can be stimulated with concurrent frontal lobe coverage is exceedingly rare. We believe
this anecdotal evidence provides further support for our claims, and we appreciate the suggestion
to include it.

*The added text regarding this can be found in the supplementary materials (Figure 2) and lines*
*235-237 of the manuscript.*

**Comment 2.9:**

The authors show convincingly that these frontal regions demonstrate similar spectrotemporal
sensitivity as regions in the lateral STG. These results would benefit by an analysis of responses
to simple tones, clicks, or white noise bursts (if any of those data were collected for these
patients). This would show whether the frontal responses are selective to speech and enable more
reliable estimates of timing in the two regions.

**Response 2.9:**

We agree that this is an important point, and we thank the Reviewer for the suggestion. Due to
the time constraints for data collection, our standard protocol is focused on presenting speech
sounds, and it is less common for us to collect such data that could speak to questions about
speech specificity. In specific cases, we have presented other types of sounds including tones and
music, however these patients typically do not have frontal lobe coverage.

*We now note in the revised manuscript (lines 290 - 293) that the focus on speech stimuli*
*preclude claims about speech specificity, and that future work should investigate this important*
*question.*

**Reviewer 3 Comments and Responses**

**Comment 3.1:**

1. What are the noteworthy results?
461 A) Current theories largely support the principle that frontal cortical responses to speech arise
primarily from sequential hierarchical speech feature encoding.
B) This study provides strong evidence supporting an innovative hypothesis that short-latency
(<200 ms) frontal cortical responses in humans are driven by parallel ascending auditory

pathways from the auditory thalamus and primary auditory cortex.
C) The dual parallel and hierarchical pathway organization for speech and other sound
processing has strong potential to radically change our computational models, simulations, and
understanding of how speech and sensory inputs modulate high-level frontal cortical processes.

**Response 3.1:**

We appreciate the Reviewer's accurate assessment of the noteworthy results.

**Comment 3.2:**

475 A) This study provides strong evidence supporting the hypothesis that short-latency (<200 ms)
frontal cortical responses in humans are driven by parallel ascending auditory pathways from the
auditory thalamus and primary auditory cortex.

B) ECoG provides high spatial and temporal resolution metrics of frontal, superior temporal
gyrus (STG), and primary auditory cortical (PAC) neural activity.

C) They find that "lower level" speech-specialized superior temporal gyrus (STG) and three
separate higher level prefrontal cortical areas (mPreCG, vSMC and IFG) respond to speech onset
with similarly short latencies (<200 ms). This supports prior work demonstrating short latency
auditory evoked ECoG responses in prefrontal cortices in primates (Romanski & Hwang
Neurosc 2012; Marrouch et al., Annals of Mathematics and Artificial Intelligence 2020).

D) Importantly, it supports the general principle that prefrontal cortical areas receive parallel
ascending thalamic and primary auditory cortical input in addition to sequential auditory cortical
pathway inputs.

E) They find that short-latency speech-driven Spectral-Temporal Response Fields (STRFs) are
similar within the speech-specialized superior temporal gyrus (STG) as well as three prefrontal
cortical areas (mPreCG, vSMC, and IFG).

F) Importantly, the speech-driven STRF responses in prefrontal and STG areas both have neural
responses that are predictive of the spectral and temporal modulations in speech sounds.

G) They find that STRF spectrotemporal encoding models predict spectral and temporal features
in speech better than semantic encoding models, for short-onset latency sites in the frontal lobe
and STG.

H) Finally, white matter tractography and functional magnetic resonance imaging (fMRI) metrics
provide evidence of parallel anatomic and functional neural pathway connections between the
auditory thalamus and the three frontal cortical areas (mPreCG, vSMC, and IFG) in addition to
primary auditory cortex on Heschl's gyrus.

This work will be significant for general auditory and speech processing fields and potentially
more broadly will help define novel principles for how ascending sensory responses modulate
top-down frontal cortical processes and executive control.

**Response 3.2:**

We appreciate the Reviewer's strongly positive assessment of the work.

To address the specific point in (C), we would like to thank the reviewer for pointing out this
prior work in primates showing short latencies in primate prefrontal cortex. The goal is to define
architecture of the speech network, and we have added these citations to draw attention to prior
informative evidence of rapid prefrontal cortex responses.

*The added text regarding this can be found on lines 55-57 and 299 of the manuscript.*

**Comment 3.3:**

This work will be significant for general auditory and speech processing fields and potentially
more broadly will help define novel principles for how ascending sensory responses modulate
top-down frontal cortical processes and executive control.

**Response 3.3:**

We appreciate the Reviewer's recognition of the significance of this work for specific and more
general audiences.

**Comment 3.4:**

525 A) These ECoG recordings provide an unparalleled, high degree of spatial and temporal
resolution that is not possible with standard electroencephalogram (EEG) and
magnetoencephalography (MEG) measures.

B) Coupling this with the tractography and fMRI analysis provides strong support for their
hypothesis of parallel short-latency ascending pathway inputs to primary auditory cortices as
well as prefrontal cortices.

**Response 3.4:**

Again, we appreciate the Reviewer's recognition of the strengths of this work, especially as it
compares to existing literature. We note that to further help situate our results, we have added
text on *lines 264-275* of the manuscript in response to other comments, which we believe also
helps to provide important context from previous work.

**Comment 3.5:**

539 A) This work is original in that it clearly demonstrates similar speech-evoked short-latency
responses in frontal and STG areas with high spatial and temporal resolution afforded by
intracranial ECoG.

B) Though they cite Romanski & Hwang Neurosc 2012 for demonstrating short latency auditory
evoked responses in prefrontal cortices in primates, they could also cite (Marrouch et al., Annals
of Mathematics and Artificial Intelligence 2020) who demonstrate similarly short latency
auditory evoked ECoG responses in primates (Fig. 4).

**Response 3.5:**

We thank the Reviewer for drawing attention to this important work. We have added this citation
to draw attention to prior informative work on rapid prefrontal cortex responses.

*The added text regarding this can be found in the discussion section on lines 55-57 and 299 of*
*the manuscript.*

**Comment 3.6:**

555 A) This work provides strong support for their general conclusions—no additions are needed.

B) In the Discussion, they may want to briefly explain that future studies could compare speech-
related features encoded in the short-latency versus longer latency responding sub-columns
within prefrontal and STG areas which they touch upon.

**Response 3.6:**

We thank the Reviewer for the suggestions. We hope this work will be further validated and built
upon and this is a helpful suggestion for a high yield future direction, and we have added text
about this point to the Discussion.

*The added text regarding this can be found in the discussion session on lines 284-287 of the*
*manuscript.*

**Comment 3.7:**

7. Are there any flaws in the data analysis, interpretation and conclusions?

570 A) No major flaws are evident.

8. Do these prohibit publication or require revision?

This manuscript reads well for a general and specialized audiences –I think it is ready to go!

9. Is the methodology sound?

576 A) Yes, the methods are sound, strong and relevant for addressing this novel hypothesis.

10. Does the work meet the expected standards in your field?

579 A) This meets high standards in the field. The same collaborators are experts in this area with
580 publications demonstrating regional topography and specialization for speech-processing within
581 STG using ECoG, STRFs, and MTF analytic techniques (Hullett, P. W., Hamilton, L. S.,
Mesgarani, N., Schreiner, C. E. & Chang, E. F. Human superior temporal gyrus organization of
spectrotemporal modulation tuning derived from speech stimuli. J
Neurosci <https://pubmed.ncbi.nlm.nih.gov/26865624/>)

11. Is there enough detail provided in the methods for the work to be reproduced?

587 A) Yes, there is ample detail in the Methods to reproduce this work.

**Response 3.7**

We greatly appreciate these positive assessments.

**Comment 3.8:**

Reviewer #3 (Remarks on code availability): I logged into github and went to the
github URL however, it says, "This repository is empty."

**Response 3.8:**

Thank you for pointing this out. The code has now been added to:

https://github.com/ChangLabUcsf/Hullett_2025

The source data has been added to the following:

<https://ucsf.box.com/s/a4o2h8yrtn5o3vgfhmjd0bz2xf7un3oi>

References

1. Manning, J. R., Jacobs, J., Fried, I. & Kahana, M. J. Broadband shifts in local field potential power spectra are correlated with single-neuron spiking in humans. *J. Neurosci.* **29**, 13613–13620 (2009).
2. Ray, S. & Maunsell, J. H. R. Different origins of gamma rhythm and high-gamma activity in macaque visual cortex. *PLoS Biol.* **9**, e1000610 (2011).
3. Steinschneider, M., Fishman, Y. I. & Arezzo, J. C. Spectrotemporal analysis of evoked and induced electroencephalographic responses in primary auditory cortex (A1) of the awake monkey. *Cereb. Cortex* **18**, 610–25 (2008).
4. Leonard, M. K. *et al.* Large-scale single-neuron speech sound encoding across the depth of human cortex. *Nature* **626**, 593–602 (2023).
5. Nourski, K. V. *et al.* Functional organization of human auditory cortex: Investigation of response latencies through direct recordings. *Neuroimage* **101**, 598–609 (2014).
6. Kajikawa, Y. & Schroeder, C. E. How local is the local field potential? *Neuron* **72**, 847–858 (2011).
7. Liu, J. & Newsome, W. T. Local field potential in cortical area MT: Stimulus tuning and behavioral correlations. *J. Neurosci.* **26**, 7779–7790 (2006).
8. Chang, E. F. Towards large-scale, human-based, mesoscopic neurotechnologies. *Neuron* **86**, 68–78 (2015).
9. Chinedu-Eneh, E. O. *et al.* Influences of electrode density on intracranial seizure localisation: a single-blinded randomised crossover study. *eBioMedicine* **113**, 105606 (2025).
10. Yeh, F., Verstynen, T. D., Wang, Y., Fernández-miranda, J. C. & Isaac, W. Deterministic Diffusion Fiber Tracking Improved by Quantitative Anisotropy. *PLoS One* **8**, 1–16 (2013).
11. Hamilton, L. S., Oganian, Y., Hall, J. & Chang, E. F. Parallel and distributed encoding of speech across human auditory cortex. *Cell* **184**, 4626–4693 (2021).